# ACVR2A facilitates trophoblast cell invasion through TCF7/c-JUN pathway in pre-eclampsia progression

**Shujing Yang[†], Huanyao Liu[†], Jieshi Hu, Binjun Chen, Wanlu An, Xuwen Song, Yi Yang\*, Fang He\***

Department of Obstetrics and Gynecology, Obstetrics; Guangdong Provincial Key Laboratory of Major Obstetric Diseases; Guangdong Provincial Clinical Research Center for Obstetrics and Gynecology; Guangdong-Hong Kong-Macao Greater Bay Area Higher Education Joint Laboratory of Maternal-Fetal Medicine; The Third Affiliated Hospital, Guangzhou Medical University, Guangzhou, China

## eLife Assessment

The role of ACVR2A is potentially of importance to both the biology of trophoblast cells and to the pathogenesis of preeclampsia. In this manuscript, the authors have taken a **useful** first step towards better understanding this protein using a loss of function model in trophoblast cell lines and then examining invasion, proliferation, and transcription in these cells. The study is **solid** and further in vivo evidence on how target factors participate in the occurrence of placental structural disorders and diseases through potential downstream pathways will be invaluable in the future.

**\*For correspondence:**
Yangyi@gzhmu.edu.cn (YY);
hefangjnu@126.com (FH)

[†]These authors contributed equally to this work

**Competing interest:** The authors declare that no competing interests exist.

**Abstract** Pre-eclampsia (PE) is a serious pregnancy disorder linked to genetic factors, particularly the ACVR2A gene, which encodes a receptor involved in the activin signaling pathway and plays a critical role in reproductive processes. Transcriptomic data analysis and experimental verification confirmed a downregulation of ACVR2A expression in placental tissues from PE patients. In this study, CRISPR/Cas9 technology was employed to investigate the effect of ACVR2A gene deletion on trophoblast cells using the HTR8/SVneo and JAR cell lines. Deletion of ACVR2A inhibits trophoblastic migration, proliferation, and invasion, underscoring its pivotal role in cellular function. RNA-seq data analysis unveiled an intricate regulatory network influenced by ACVR2A gene knockout, especially in the TCF7/c-JUN pathway. By employing RT-PCR and immunohistochemical analysis, a potential association between ACVR2A and the TCF7/c-JUN pathway was hypothesized and confirmed. The complexity of PE onset and the significance of genetic factors were emphasized, particularly the role of the ACVR2A gene identified in genome-wide association study. This study established a robust foundation for delving deeper into the intricate mechanisms of PE, paving the way for focused early intervention, personalized treatment, and enhanced obstetric healthcare.

## Introduction

Pre-eclampsia (PE) is a pregnancy-related disorder characterized by the onset of hypertension and proteinuria typically occurring after the 20th week of gestation (*Chen et al., 2020*). The clinical manifestations of PE range from asymptomatic high blood pressure detected during routine prenatal examinations to severe symptoms such as seizures, respiratory distress, upper abdominal discomfort, and evident placental abruption (*Dimitriadis et al., 2023*). Genetic factors are crucial in PE pathogenesis, with a positive family history being a significant risk factor (*Chappell et al., 2021*). Although the

exact mechanisms of PE remain unclear, with suggested mechanisms including endothelial damage, excessive immune activation, nutritional deficiencies, genetic factors, and insulin resistance (*Chappell et al., 2021*; *Gebara et al., 2021*). The insufficient infiltration of trophoblasts, resulting in defective remodeling of uterine spiral arteries, widely recognized as a pivotal contributor to the pathogenesis of PE (*Knöfler et al., 2019*). Additionally, dysregulated immune responses, abnormal activation of the renin–angiotensin system activity, chronic inflammation, and platelet hyperactivation have been implicated in the progression of this condition (*Ferreira et al., 2017*; *Daussin et al., 2008*). Understanding the complex between these factors is essential for unraveling the mechanisms driving PE onset and progression and for developing effective therapeutic strategies (*Dimitriadis et al., 2023*).

Genome-wide association studies (GWASs) have significantly enhanced the understanding of the genetic underpinnings of complex diseases, including pregnancy complications such as PE, in recent years. By analyzing single-nucleotide polymorphisms (SNPs) across millions of genetic loci in thousands of individuals, GWASs have successfully identified genetic markers associated with disease susceptibility (*Hua, 2023*). In the context of PE, GWASs has uncovered multiple risk loci linked to key biological processes, such as vascular development, blood pressure regulation, and immune response (*Li et al., 2022*). However, the genetic architecture of PE is highly complex, and the contribution of individual genetic variation is generally modest. Moreover, GWAS findings often require functional studies to validate their biological significance and to decipher the underlying pathological mechanisms. For example, identifying how specific risk loci, such as those associated with the ACVR2A gene, influence trophoblast behavior and placental function is essential for translating genetic insights into clinical applications.

Recent GWASs have highlighted the pivotal role of the ACVR2A gene in the development of hypertensive disorders of pregnancy (HDP) and PE (*Yong et al., 2018a*). Studies conducted across diverse populations, including the Philippines, Northeast Brazil, Australia/New Zealand, Norway, and China, have consistently demonstrated significant associations between ACVR2A gene polymorphisms and PE susceptibility (*Ferreira et al., 2015*; *Yanan et al., 2020*; *Roten et al., 2009*; *Fitzpatrick et al., 2009*; *Amosco et al., 2019*; *Moses et al., 2006*; *Zeybek et al., 2013*; *Glotov et al., 2019*). Located on chromosome 2q22, ACVR2A is a critical member of the receptor family mediating activin function within the transforming growth factor-β (TGF-β) ligand family, exerting diverse biological functions (*Dean et al., 2017*). Elevated levels of activin A, primarily secreted by the placenta, are well documented in the progression of HDP. ACVR2A plays an indispensable role in promoting endometrial decidualization and regulating the growth and development of the placenta and embryo (*Yong et al., 2015*). As a receptor for activin A, ACVR2A is crucial in reproductive processes, including decidualization, trophoblast invasion, and placental formation during pregnancy (*Glotov et al., 2019*). In addition to activin A, ACVR2A is also a receptor for other members of the TGF-β superfamily, including bone morphogenetic proteins (BMPs) and other activins, with varying affinities and coreceptors. For instance, BMP2 has been shown to stimulate trophoblast invasion, at least in part via ACVR2A, suggesting a broader role for ACVR2A in trophoblast regulation (*Zhao et al., 2018*). These diverse interactions highlight the complexity of ACVR2A-mediated signaling in trophoblast differentiation and invasion, which merits further exploration. The receptor's predominant expression in the endometrium, placenta, and vascular endothelial cells underscores its core role in pregnancy-related mechanisms (*Lokki et al., 2011*). Collectively, these findings support the hypothesis that ACVR2A gene polymorphisms may disrupt placental formation, contributing to early-onset PE. Future studies are warranted to elucidate the precise molecular mechanisms underlying ACVR2A dysfunction in PE and its potential as a therapeutic target.

## Results

### Downregulated expression of ACVR2A in PE tissues reveals a significant correlation

Several studies have established significant correlations between ACVR2A gene polymorphisms and the susceptibility to PE in different populations. *Table 1* summarizes the key findings from these studies, highlighting the associations between various SNPs in the ACVR2A gene and the incidence of PE. These correlations suggest that genetic variations in ACVR2A may play a crucial role in the pathogenesis of PE.

**Table 1.** ACVR2A gene polymorphisms and pre-eclampsia risk summary.

| Study participants | Associated SNPs | p-value | OR | Nature of variant (s) | Reference |
|---|---|---|---|---|---|
| | rs1014064 | 0.556 | 0.8699 | Intronic | |
| | rs2161983 | 0.4717 | 0.8921 | Intronic | |
| 150 PE cases and 175 controls; Philippine women | Rs1014064, Age, BMI | 0.005 | | Intronic | *Amosco et al., 2019* |
| | rs1424954 | 0.002 | 1.86 | Promoter region | |
| | rs1014064 | 0.004 | 1.77 | Intronic | |
| 443 PE cases and 693 controls; Brazil women; association with early-onset pre-eclampsia found after grouping in accordance to gestational age at delivery | rs2161983 | 0.008 | 1.70 | Intronic | *Ferreira et al., 2015* |
| | rs3768687 | 0.039 | 1.52 | Intronic | |
| | rs10497025 | 0.025 | | Intronic | |
| | rs13430086 | 0.010 | | 3'UTR | |
| | LF004 | 0.018 | | Intronic | |
| | LF013 | 0.018 | | Intronic | |
| 176 PE cases, 20 eclampsia and 90 controls; Australian/New Zealand women | LF020 | 0.018 | | Intronic | *Fitzpatrick et al., 2009* |
| | rs1014064 | 0.0184 | 0.86 | Intronic | |
| | rs17742134 | 0.0214 | 1.17 | Intronic | |
| | rs1424941 | 0.0171 | 1.18 | Intronic | |
| | rs2161983 | 0.0196 | 0.86 | Intronic | |
| | rs3768687 | 0.0214 | 0.86 | Intronic | |
| 1139 PE cases and 2269 controls; Norwegian women | rs3764955 | 0.0327 | 0.87 | Intronic | *Roten et al., 2009* |
| | rs1424954 | 0.007 | | Promoter region | |
| | rs1364658 | 0.04 | | Intronic | |
| 121 PE cases and 71 controls; Australian/New Zealand women | rs1895694 | 0.05 | | Intronic | *Moses et al., 2006* |
| | rs1424954 | 0.013 | 0.687 | Promoter region | |
| | rs1014064 | 0.016 | 0.693 | Intronic | |
| | rs1128919 | 0.018 | 0.536 | Synonymous | |
| | rs3768687 | 0.019 | 0.701 | Intronic | |
| | rs3764955 | 0.024 | 1.784 | Intronic | |
| 140 PE cases and 380 controls; Northern Chinese women | rs13430086 | 0.029 | 0.729 | 3'UTR | *Yanan et al., 2020* |
| | rs1128919 | 0.02 | 0.44 | Synonymous | |
| | rs13430086 | 0.02 | 0.28 | 3'UTR | |
| 94 PE cases and 116 controls; Turkish women | rs10497025 | 0.025 | 0.010 | Intronic | *Zeybek et al., 2013* |

These findings underscore the significance of ACVR2A gene variants in the initiation of PE. In this study, placental samples were obtained from 40 patients, and the detailed clinical attributes are presented in *Supplementary file 1*. No significant differences in age and pre-pregnancy BMI were observed between the PE group and the normal control (NC) group (p > 0.05). However, the average birth weight of infants delivered by mothers in the PE group was markedly lower than that in the control group (p < 0.05).

Analysis of transcriptome data from a public database (GSE114691) confirmed a marked reduction in ACVR2A expression in the PE group, consistent with its potential involvement in impaired placental function (*Figure 1A*). *Figure 1B* shows a heatmap of differentially expressed genes, with a focus on those enriched in the WNT signaling pathway. These genes were identified based on significant

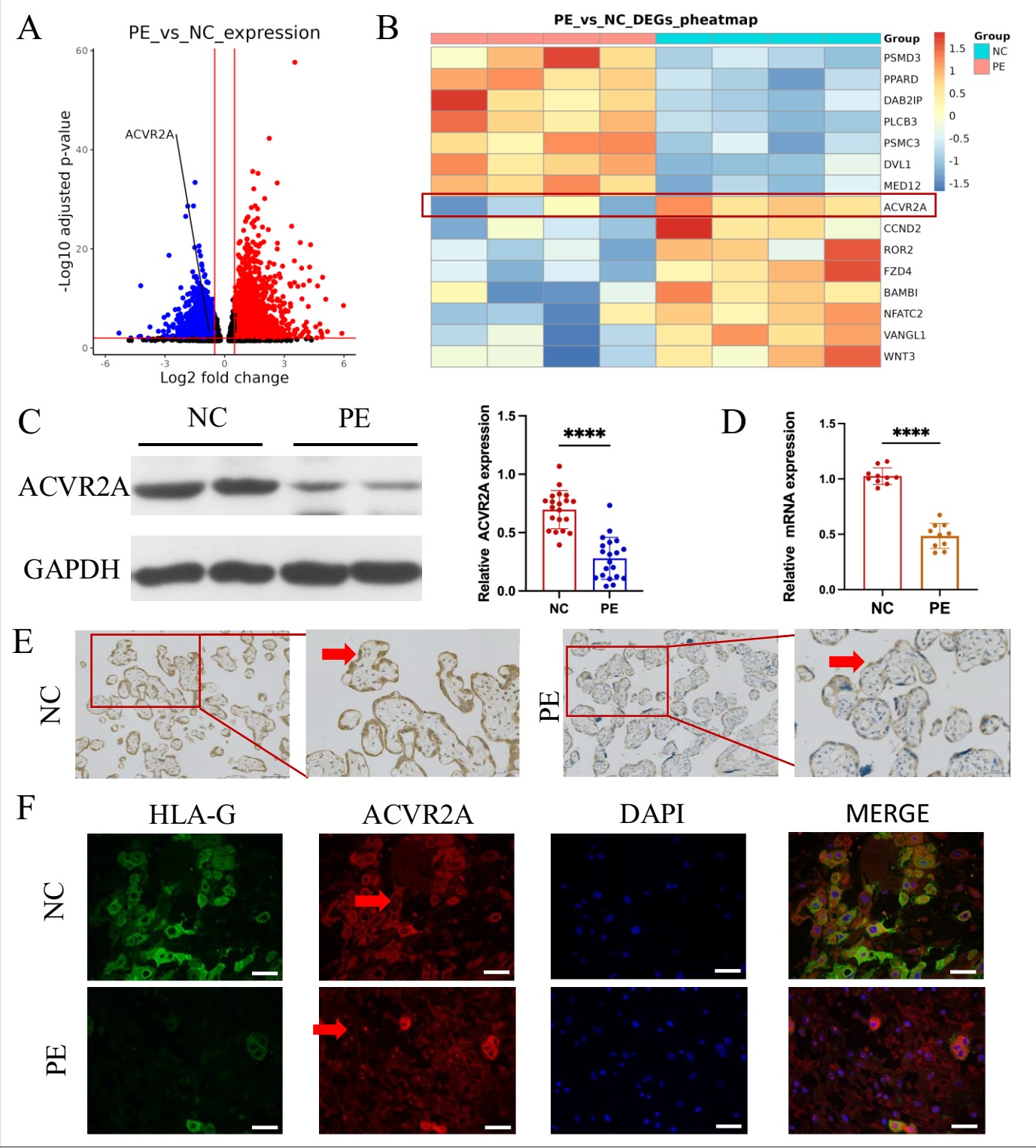

**Figure 1.** ACVR2A was downregulated in placental tissue associated with pre-eclampsia (PE). (**A**) Comparison of the RNA-seq volcano maps of all genes in the placenta of normal control (NC) and patients with PE evidently showed that the expression of the ACVR2A gene significantly decreased in patients with PE. (**B**) Heatmap of differentially expressed genes in PE versus control placental samples. Genes were selected based on significant differential expression (false discovery rate [FDR] <0.05, |log₂(fold change)| ≥1) and their involvement in the WNT signaling pathway, as determined through KEGG and GSEA analyses. ACVR2A is highlighted alongside other genes of interest to illustrate its association with the molecular landscape of PE. (**C**) Western blot analysis demonstrated reduced levels of ACVR2A in PE placental tissue (*n* = 20) compared with control placentas. (**D**) RT-qPCR was employed to assess the ACVR2A mRNA expression in placental tissues of NC (*n* = 10) and patients with PE (*n* = 10). (**E**) Immunohistochemical

*Figure 1 continued on next page*

*Figure 1 continued*

staining was conducted using rabbit IgG anti-human ACVR2A antibody on sections from NC and PE placentas. Sections were counterstained with hematoxylin. The ACVR2A levels were markedly lower in patients with PE (*n* = 10) than in NC. (**F**) Immunofluorescence co-localization of rabbit IgG anti-human ACVR2A antibodies and HLA-G antibodies (a marker of extrachorionic trophoblastic cells) was performed in NC and PE maternal placenta. The expression pattern of the ACVR2A antibody closely resembles that of the HLA-G antibody, primarily expressed in extravillous trophoblast (EVT) cells (****p < 0.0001 compared with NC group).

The online version of this article includes the following source data and figure supplement(s) for figure 1:

**Source data 1.** Original western blot shown in *Figure 1C* (labelled).

**Source data 2.** Original western blot shown in *Figure 1C*.

**Figure supplement 1.** Downregulation of ACVR2A expression in pre-eclampsia placental tissue: insights from western blot and immunohistochemistry.

**Figure supplement 1—source data 1.** Original gel image of ACVR2A shown in *Figure 1—figure supplement 1A* (labelled).

**Figure supplement 1—source data 2.** Original gel image of ACVR2A shown in *Figure 1—figure supplement 1A*.

**Figure supplement 2.** Reduced ACVR2A expression in trophoblast cells of pre-eclampsia placental tissue revealed by immunofluorescence co-localization with HLA-G.

differential expression and pathway enrichment analyses using KEGG and Gene Set Enrichment Analysis (GSEA).

Western blot and RT-PCR analyses validated these findings, showing significantly reduced ACVR2A protein and mRNA levels in PE placental samples compared to controls (p < 0.001, *Figure 1C, D*, *Figure 1—figure supplement 1A*). Immunohistochemical (IHC) staining revealed lower signal intensity of ACVR2A in PE placentas compared to NC placentas, with positive staining observed in both extravillous trophoblasts (EVTs) and syncytiotrophoblasts. In *Figure 1E*, EVTs are indicated with arrows in the higher magnification image to improve clarity, and additional examples are provided in *Figure 1—figure supplement 1*. Experimental evidence has indicated predominant expression of HLA-G in trophoblast cells of EVTs, a critical site at the maternal–fetal interface, where trophoblast cells closely interact with the maternal uterine wall (*Ferreira et al., 2017*). Consequently, immunofluorescence focused on the placental maternal surface, with concurrent staining for ACVR2A and HLA-G. Immunofluorescence analysis revealed a diminished signal of ACVR2A in PE placentas compared to NC placentas, with predominant expression observed in trophoblast cells (*Figure 1F*, *Figure 1—figure supplement 2*). To provide additional context, lower magnification images showing the location of the anchoring villi are included in *Figure 1—figure supplement 2*, while *Figure 1F* highlights ACVR2A staining in EVTs at higher magnification to better visualize its expression patterns. ACVR2A was primarily localized to the cell surface, consistent with its role as a membrane receptor.

## Precision genome surgery: ACVR2A knockout via CRISPR/Cas9

Preliminary experiments confirmed the expression of ACVR2A in trophoblast cells. The expression of ACVR2A in trophoblast cells was verified through RT-qPCR conducted in our laboratory, as shown in *Figure 2A*. Data from NCBI and the Human Protein Atlas were used to support our findings and provide broader context regarding ACVR2A expression patterns across different cell types. The mRNA levels were analyzed using RT-PCR in various cell types, including HTR8/SVneo, JAR, HCoEpiC (human normal colon epithelial cells), 293 human embryonic kidney cells, MEG-01 (human megakaroblastic leukemia cells), Huh-7 (human liver cancer cells), NCI-H358 (human non-small cell lung cancer cells), HaCaT (human immortal epidermal cells), induced pluripotent stem (iPS) cells, A427 (human lung cancer cells), and A549 (human non-small cell lung cancer cells), to understand the expression of ACVR2A in different cell lines. In *Figure 2A*, the mRNA levels of 293 and iPS cells were lower than those of the other cell lines, whereas the mRNA expression levels of HTR8/SVneo and JAR cells resembled those of other cancer cells. Notably, the mRNA expression of JAR cells was significantly higher than that of the other cancer cell lines.

We hypothesized a potential association between ACVR2A expression and trophoblast cell behavior. To test this, we compared ACVR2A mRNA expression levels across various cell lines, including trophoblast-derived (HTR8/SVneo, JAR) and non-trophoblast cell lines (e.g., A549, Huh7). As shown in *Figure 2A*, ACVR2A expression was significantly higher in trophoblast-derived cell lines, particularly in JAR cells, highlighting its potential role in trophoblast-specific functions such as invasion and migration. In addition, ACVR2A expression in placental tissues from PE patients and normotensive controls

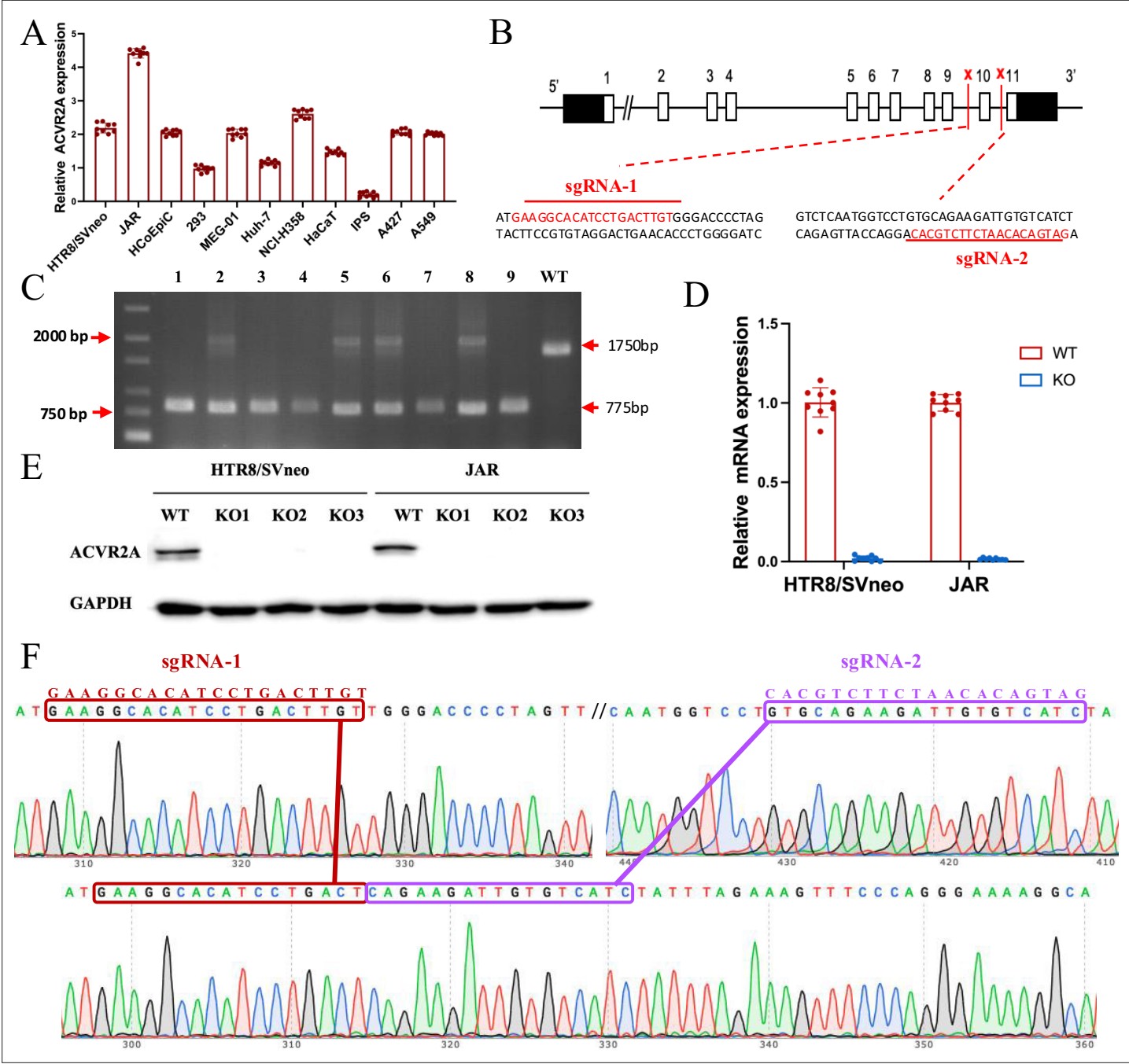

**Figure 2.** The ACVR2A gene in JAR and HTR8/SVneo cells was successfully knocked out using CRISPR/Cas9 gene editing technology. (**A**) RT-qPCR was employed to assess the ACVR2A mRNA expression across various cell lines. The expression of ACVR2A in HTR8/SVneo and JAR was comparable to that in multiple cancer cell lines, with JAR exhibiting a higher ACVR2A expression. (**B**) DNA sequence of ACVR2A and target sequence site information of sgRNA-1 and sgRNA-2. (**C**) The knockout efficiency of the ACVR2A gene in HTR8/SVneo and JAR cell lines was assessed through polymeric primer PCR, confirmed by agarose gel electrophoresis, and reconfirmed through four rounds of monoclonal cell communities. The specific experimental results are shown in *Figure 2—figure supplement 1*. (**D**) The ACVR2A mRNA levels were measured in HTR8/SVneo and JAR cell lines, and the normal control group was compared with the two ACVR2A-KO cell lines after three rounds of validation . (**E**) Western blot analysis showing ACVR2A protein levels in wild-type (WT) and knockout (KO) HTR8/SVneo and JAR cell lines. The ACVR2A protein is significantly absent in KO cell lines compared to WT, confirming successful knockout of the ACVR2A gene. (**F**) Sanger sequencing confirmed the expression of ACVR2A in ACVR2A-KO monoclonal cell lines and successfully knocked out the DNA fragment between the ACVR2A gene sgRNA-1 and sgRNA-2. The specific experimental results are shown in *Figure 2—figure supplements 2 and 3*.

The online version of this article includes the following source data and figure supplement(s) for figure 2:

*Figure 2 continued on next page*

*Figure 2 continued*

**Source data 1.** Original western blot shown in *Figure 2E* (labelled).

**Source data 2.** Original western blot shown in *Figure 2E*.

**Figure supplement 1.** Genotype identification led to the selection of three pairs of HTR8/SVneo and JAR cell lines with complete ACVR2A knockout.

**Figure supplement 2.** The complete knockout of the ACVR2A gene in HTR8/Svneo monoclonal cell lines was confirmed through Sanger sequencing electrophoresis.

**Figure supplement 3.** The complete knockout of the ACVR2A gene in JAR monoclonal cell lines was confirmed through Sanger sequencing electrophoresis.

**Figure supplement 1—source data 1.** Original agarose gel electrophoresis gel patterns shown in *Figure 2—figure supplement 1* (labelled).

**Figure supplement 1—source data 2.** Original agarose gel electrophoresis gel patterns shown in *Figure 2—figure supplement 1*.

showed a marked reduction in PE placentas. Together, these findings underscore the importance of ACVR2A in trophoblast behavior and its potential contribution to PE pathogenesis.

This study focused on the HTR8/SVneo and JAR cell lines, both derived from human EVT cell lines during early pregnancy. Originating from chorionic villi in early pregnancy, the HTR-8/SVneo cell line serves as a prevalent in vitro model for cultured human cells. Researchers frequently opt for the HTR8/SVneo cell line as an alternative model to investigate placental function and study pregnancy-related diseases.

JAR cell lines derived from placental tissue exhibited heightened sensitivity to morphological and biological changes induced by trophoblastic-specific factors (*Zeng et al., 2018*). Given their choriocarcinoma properties, JAR cells offer a valuable model for simulating and studying trophoblast invasion during pregnancy (*Alvarez-Cienfuegos et al., 2020*). Thus, JAR and HTR8/SVneo cell lines constituted the cell models employed in the present study.

We employed CRISPR/Cas9 technology to delete the ACVR2A gene in HTR8/SVneo and JAR cell lines. The successful knockout was confirmed through Sanger sequencing, DNA agarose gel electrophoresis, and RT-PCR. This allowed us to investigate the effect of ACVR2A deletion on trophoblast cell behavior (*Jinek et al., 2012*). By employing the CRISPR/Cas9 conditional knockout system, two single-guide RNAs (sgRNAs) were designed to induce the knockout of the ACVR2A gene in two cell lines through electroporation (*Figure 2B*). The genotypic fragment resulting from the double sgRNA knockout was 775 bp, and the genotypic fragment of the wild type (WT) measured 1750 bp (*Figure 2—figure supplement 1*). To establish ACVR2A double-knockout cell lines, CRISPR/Cas9 genome editing was performed, followed by monoclonal culture and rigorous screening. Genotyping through PCR amplification and sequencing confirmed the successful incorporation of indels at the ACVR2A target loci, identifying specific monoclonal clones with double-allele mutations. RT-qPCR and western blotting further validated the absence of ACVR2A expression at both transcript and protein levels in these clones, confirming the successful establishment of ACVR2A double-knockout monoclonal cell lines for downstream functional studies (*Figure 2C–E*, $p < 0.001$). Subsequent Sanger sequencing confirmed this outcome, revealing the successful knockout of the gene segment between the two sgRNAs (*Figure 2F*, *Figure 2—figure supplements 2 and 3*). These experiments helped further investigate the complex process of ACVR2A's involvement in trophoblast function and reveal its potential regulatory role in trophoblast proliferation and invasion. The use of advanced gene manipulation techniques, such as CRISPR/Cas9, ensured the accuracy and validity of the experimental approach, ensuring the successful deletion of ACVR2A in the target cell line. This provides a foundation for further study of the downstream effects and molecular mechanisms of ACVR2A on trophoblastic function.

## ACVR2A knockout diminishes in vitro cell migration and invasion

Cells at the periphery of the scratch could progressively migrate into the void, facilitating the healing of the scratch. The knockout of ACVR2A markedly impeded the migration of HTR8/SVneo and JAR cells in comparison to NCs ($p < 0.001$, *Figure 3A*). As illustrated in *Figure 3B*, the ACVR2A gene knockout markedly suppressed the proliferation rate of HTR8/SVneo cells and JAR cells in comparison to NC ($p < 0.001$, *Figure 3B*). Transwell invasion assays further demonstrated reduced invasive capabilities ($p < 0.001$, *Figure 3C*). Colony formation assays revealed a significant decrease in clonogenicity in ACVR2A knockout cells ($p < 0.001$, *Figure 3D*). The experiments established that ACVR2A

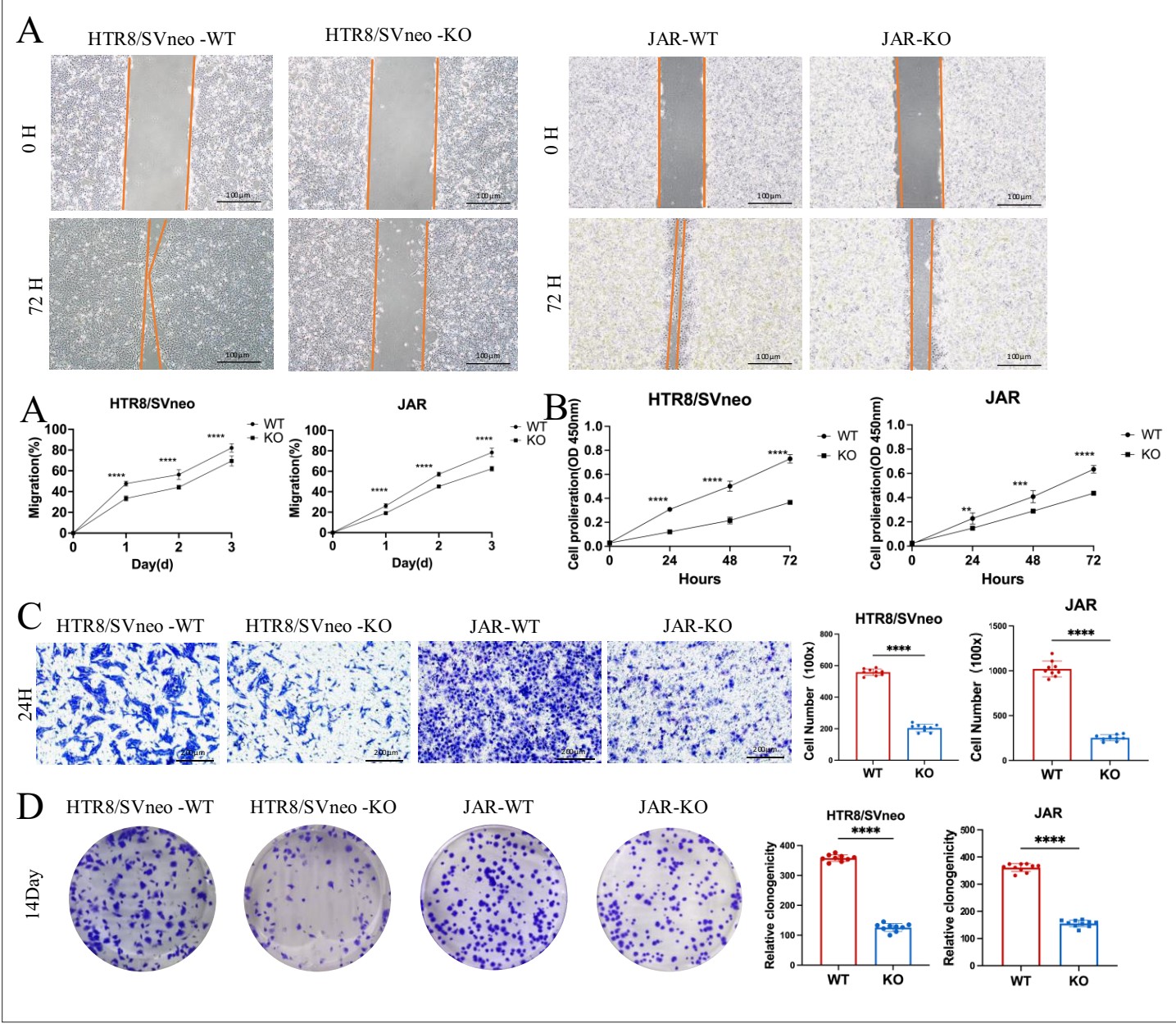

**Figure 3.** Effects of ACVR2A knockout on trophoblast cell function. (**A**) Cell scratch assay was conducted in a 6-well plate to assess alterations in the migration ability of HTR8/SVneo and JAR cells following ACVR2A gene knockout. Three visual fields were randomly selected under a 100×microscope for continuous observation, calculation, and difference analysis. Scale bar: 100 μm. (**B**) CCK-8 method was employed to assess the proliferation of HTR8/ SVneo and JAR cells after ACVR2A gene knockout. (**C**) Cell invasion was quantified by counting cells in five randomly selected fields of view at ×300 magnification. Images shown in the figure were captured at lower magnification (×100) to provide an overview of the experimental and control groups. Scale bar: 200 μm. (**D**) Colony formation was detected by single-cell clone assay. Colony formation assay conducted in a 6-well plate to assess changes in the individual cell proliferation capacity of HTR8/SVneo and JAR cells following ACVR2A gene knockout. Colony formation assays were repeated independently at least 10 times to ensure reproducibility. The results are expressed as the mean ± SD of these replicate experiments (*p < 0.05, **p < 0.01, ***p < 0.001, and ****p < 0.0001 compared with normal control group).

plays a crucial role in regulating fundamental cell functions, and its knockout markedly hinders cell migration, proliferation, invasion, and clonal formation. The results hold important implications for comprehending the role of ACVR2A in cellular biology and the development of diseases.

## RNA-seq unveils ACVR2A-mediated regulation of trophoblast cell migration and invasion via Wnt pathway

Transcriptome sequencing, also known as RNA-seq, is the latest sequencing technology for comprehensive and rapid transcriptome analysis, providing gene sequence and transcriptomic data for specific cells or tissues of a given species. ACVR2A has been shown to regulate the expression of numerous genes in its proximity. Comprehensive RNA sequencing was performed on both ACVR2A knockout (ACVR2A-KO) and WT cells to uncover the intricate mechanisms through which ACVR2A regulates HTR8/SVneo and JAR cell function. In the JAR knockout group, the deletion of ACVR2A led to the upregulation of 144 genes and the downregulation of 240 genes. By contrast, in the HTR8/SVneo cells, 99 genes were upregulated, and 73 genes were downregulated (*Figures 4A and 5A*). GSEA is a computational approach that facilitates the evaluation of the enrichment of predefined gene sets within a sequenced gene list. GSEA serves as a powerful tool that aids researchers in gaining profound insights into genomic data and elucidating crucial genes and pathways involved in biological processes.

GSEA was employed to elucidate the effect of ACVR2A gene knockout on critical genes and biological pathways in cellular systems, and the canonical pathways from the Cancer Genome Project (CGP) database were investigated. The results suggested modifications in several biological pathways. Information from the KEGG website was integrated with the analysis data of this study, uncovering substantial variations in the Wnt signaling pathway. In JAR cells, the NES value was −1.67, with a p-value of 1.03E−04, whereas in HTR8/SVneo cells, the NES value was −1.45, with a p-value of 0.6E−02 (*Figures 4B and 5B*). Dotplot and centplot visualizations were created, highlighting the connections among at least 10 pathways associated with cell migration and invasion (*Figures 4C and 5C*). These plots provide insights into the critical genes and biological pathways that ACVR2A may regulate in cellular processes. Detailed gene analysis of specific, related biological pathways is presented in *Figures 4D and 5D*. Additionally, *Figures 4E and 5E* visually represent the differentially expressed genes within the Wnt signaling pathway.

## ACVR2A regulates the migration and invasion of trophoblast cells through TCF7/c-JUN pathway

Previous experiments have confirmed the pivotal role of ACVR2A in regulating the fundamental functions of trophoblast cells. Knocking out ACVR2A significantly hindered the migration, proliferation, invasion, and clonal formation of trophoblast cells. The results of RNA-seq analysis showed that ACVR2A knockout resulted in aberrations in multiple signaling pathways, including cell migration, TGF-β, and Wnt. By integrating information from the KEGG website with the analytical data in this study, ACVR2A was identified to potentially regulate the fundamental biological functions of trophoblastic cells through the TCF7/c-JUN signaling pathway. Relevant genes enriched in the TCF7/c-JUN pathway, such as Wnt3, Wnt4, c-JUN, CCND1, TCF7, and TCF7L1, were selected for detailed investigation to further validate the effect of ACVR2A on specific gene expression. The RT-PCR results demonstrated that the mRNA expression levels at tissue and cell levels significantly decreased following ACVR2A knockout compared with the NC group (p < 0.001, *Figure 6A–C*). Additionally, IHC analysis was employed to assess the protein expression levels of TCF7, TCF7L1, and c-JUN in clinical placental samples from PE patients and NCs. As depicted in *Figure 6D–I*, the expression levels of these proteins were significantly reduced in the placental tissues from PE patients compared to NCs. Western blot analysis further validated the findings obtained from IHC analysis. As shown in *Figure 7A*, the expression levels of ACVR2A, SMAD4, SMAD1/5, and their phosphorylated forms (pSMAD1/5/9) were significantly lower in PE placentas compared to normotensive controls (*P*<0.05). Similarly, downstream targets of the TCF7/c-JUN pathway, including TCF7L1 and TCF7L2, also exhibited reduced protein expression in PE tissues. These results align with the transcriptomic data and suggest a functional impairment of the ACVR2A-TCF7/c-JUN axis in PE.

These findings support the hypothesis that reduced ACVR2A expression is associated with altered TCF7/c-JUN pathway activity in PE placental tissues. These findings further illuminate the intricate

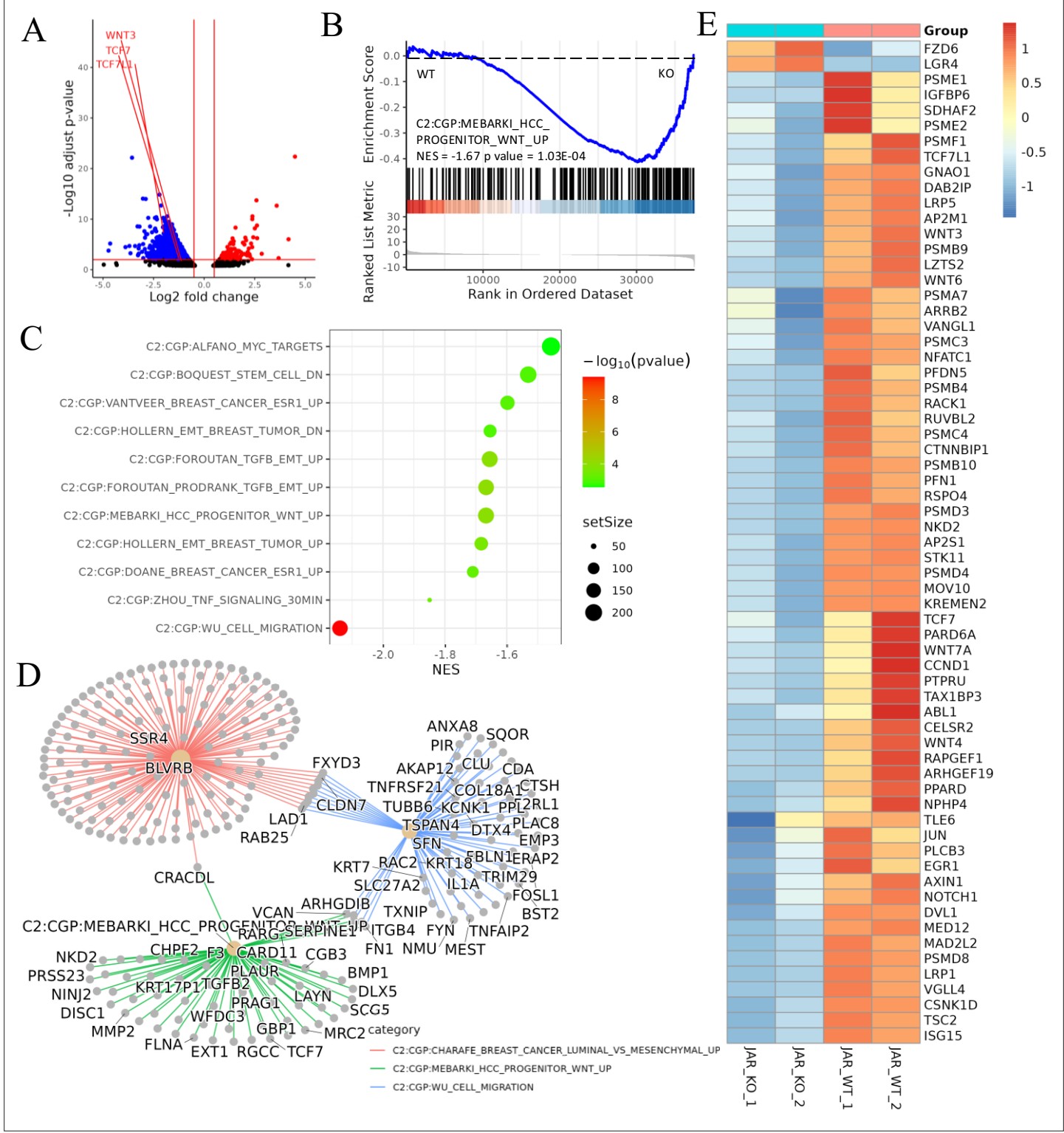

**Figure 4.** Transcriptomic analysis revealed that ACVR2A may suppress cell biological behavior through the Wnt/TCF7 pathway in JAR cell lines. (**A**) The volcano plots of RNA-seq for all the genes compared JAR-ACVR2A-KO and WT. (**B**) One representative hallmark pathway, the Wnt pathway, in the ACVR2A-KO group. (**C**) The enriched biological process pathways of cell invasion and migration based on differentially expressed genes (DEGs) (p-value cutoff = 0.05). (**D**) The category netplot depicted the linkages of downregulated genes and three biological concepts that are related to ACVR2A as a network. (**E**) The heatmap showed DEGs in the Wnt pathway between experimental and normal control groups. Red and blue represent significantly upregulated and downregulated genes, respectively.

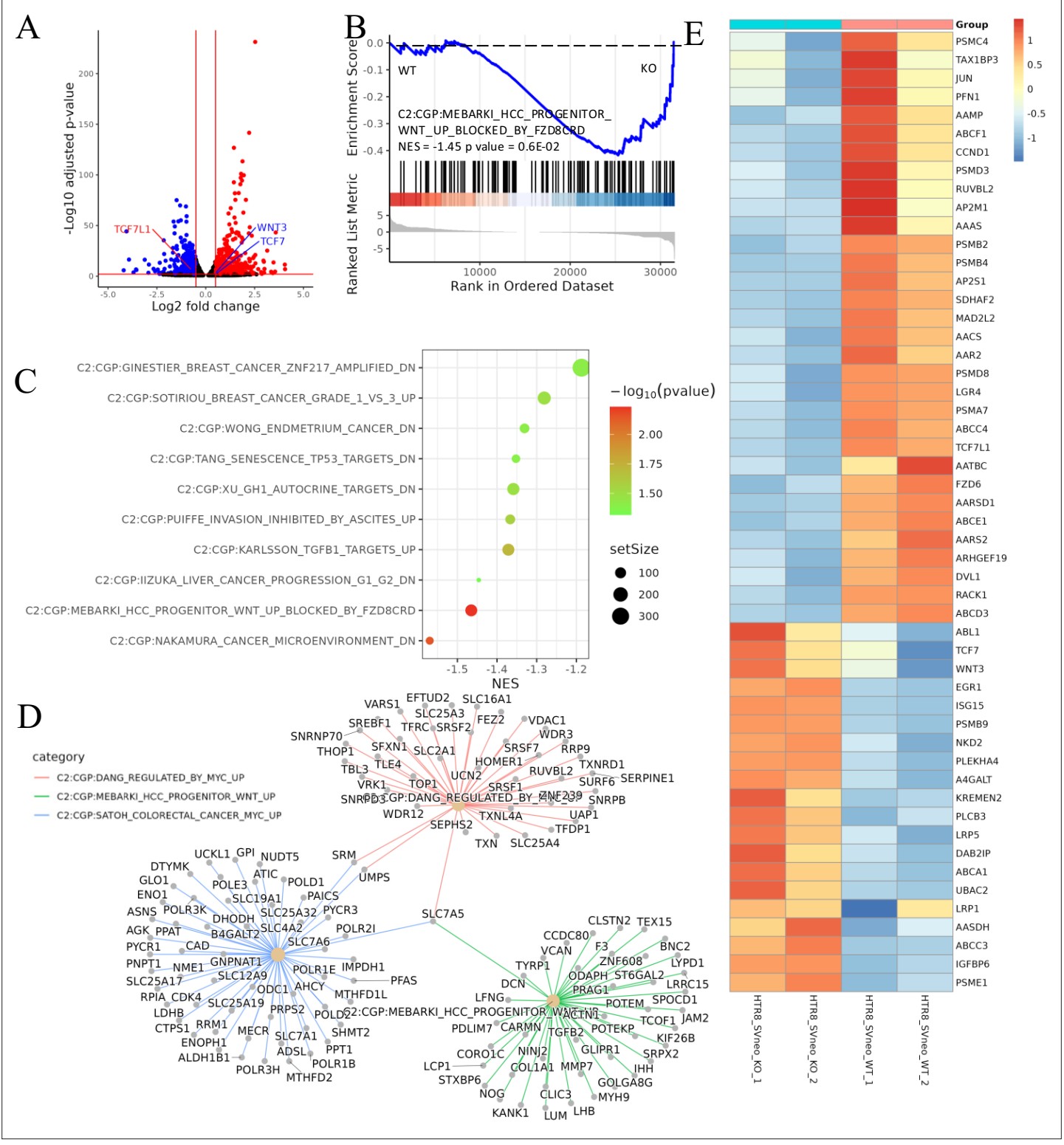

**Figure 5.** Transcriptomic analysis revealed that ACVR2A may suppress cell biological behavior through the Wnt/TCF7 pathway in HTR8/SVneo cell lines. (**A**) Volcano plots of RNA-seq for all the genes compared HTR8/SVneo-ACVR2A-KO and WT. (**B**) One representative hallmark pathway, the Wnt pathway, in the ACVR2A-KO group. (**C**) The enriched biological process pathways in cell invasion and migration based on differentially expressed genes (DEGs) (p-value cutoff = 0.05). (**D**) The category netplot depicted the linkages of downregulated genes and three biological concepts that are related to ACVR2A as a network. (**E**) Heatmap of DEGs about Wnt pathway between experimental and normal control groups. Red and blue represent significantly upregulated and downregulated genes, respectively.

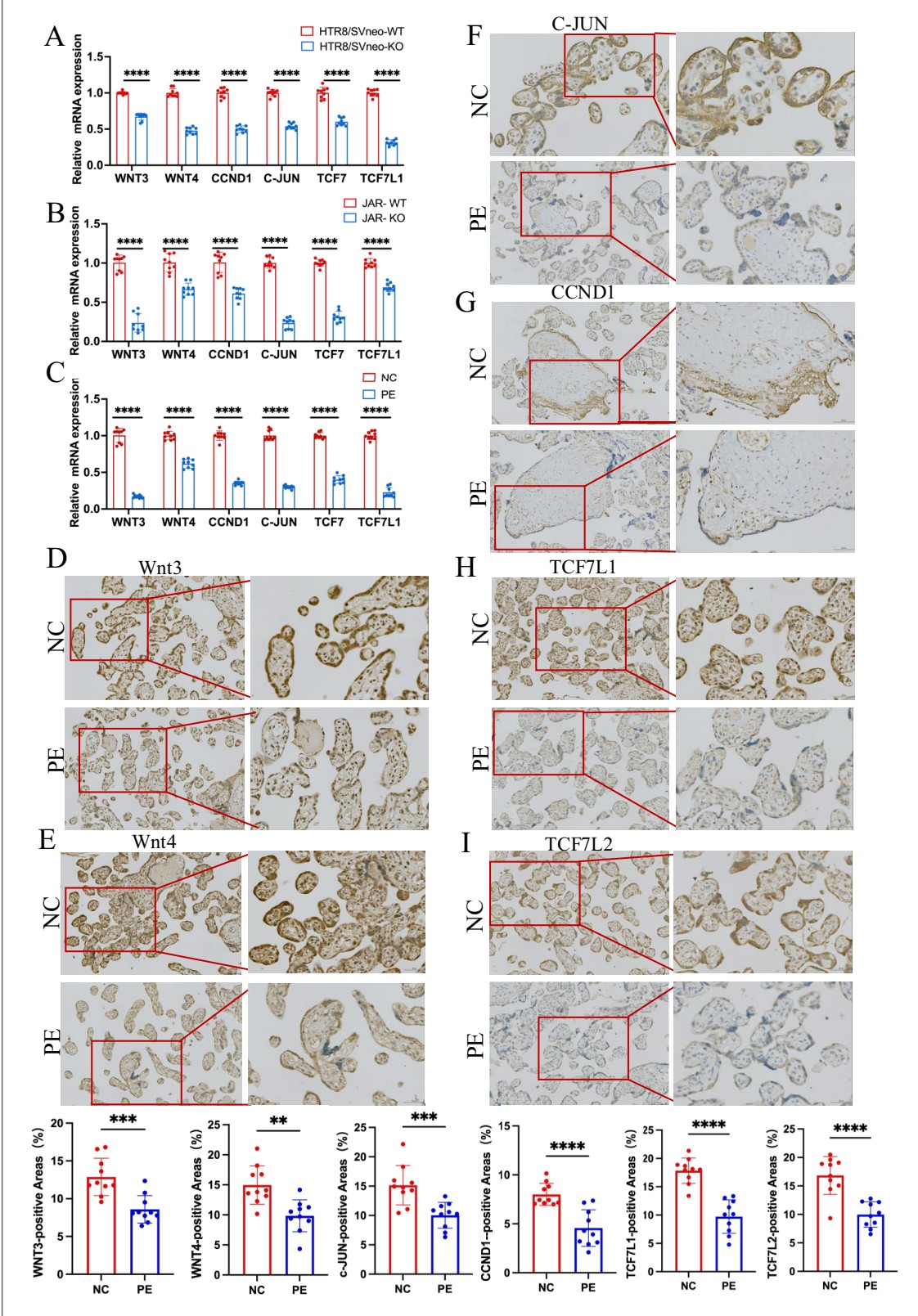

**Figure 6.** RT-PCR and immunohistochemistry validated that ACVR2A modulates cellular behavior via the TCF7/c-JUN pathway. (**A**) RT-qPCR analysis of differentially expressed genes (DEGs) (Wnt3, Wnt4, TCF7, TCF7L1, CCND1, and c-JUN) expression enriched in Wnt/TCF pathway following HTR8/SVneo ACVR2A knockout. (**B**) RT-qPCR analysis of DEGs (Wnt3, Wnt4, TCF7, TCF7L1, CCND1, and c-JUN) expression enriched in Wnt/TCF pathway following JAR ACVR2A knockout. (**C**) RT-qPCR analysis of DEGs (Wnt3, Wnt4, TCF7, TCF7L1, CCND1, and c-JUN) expression enriched in Wnt/TCF pathway

*Figure 6 continued on next page*

*Figure 6 continued*

expression in the placenta of normal control pregnant women (NC) and patients with pre-eclampsia (PE). (**D-I**) Immunohistochemical staining of normal control pregnant women and PE placentas using rabbit IgG anti-human Wnt3, Wnt4, TCF7L1, TCF7L2, CCND1, and c-JUN antibody. Sections were counterstained with hematoxylin, and positive cells were quantified using ImageJ software (**p < 0.01; ***p < 0.001;****p < 0.0001).

mechanisms by which ACVR2A regulates the biological functions of trophoblast cells. In conclusion, the diminished expression of ACVR2A significantly influences the proliferation, invasion, and migration of trophoblastic cells in women with PE.

## Discussion

PE is a severe pregnancy complication that significantly risks mothers and fetuses, driven by complex genetic and environmental interactions, and is closely linked to placental abnormalities such as defects in implantation, trophoblast dysfunction, and abnormal vascular development (*Chappell et al., 2021*; *Steegers et al., 2010*). Advanced maternal age is a notable risk factor, particularly with the trend of delayed childbearing (*Miller et al., 2022*; *Bartsch et al., 2016*; *Smithson et al., 2022*). Understanding PE mechanisms and developing suitable models are crucial but challenging due to the unique nature of human placental dysfunction (*Carter, 2011*). Trophoblast cells are vital during pregnancy, supporting embryo attachment, decidualization, vascular remodeling, hormone production, and nutrient transport. Their dysfunction is closely linked to PE, making them a key focus in PE research (*Knöfler et al., 2019*). Most studies, including ours, use trophoblast cells in in vitro and in vivo models such as cell lines, placental explants, and animal models to investigate PE mechanisms and pathogenesis.

In recent years, GWASs have achieved significant progress in elucidating the genetic basis of PE. Several risk loci associated with PE, including ACVR2A, MTHFR, and FGF, have been identified (*Tyrmi et al., 2023*; *Salonen Ros et al., 2000*; *Cnattingius et al., 2004*). In-depth exploration of these genes enhances understanding of individual variations in PE, aiding early prediction, intervention, and treatment. A comprehensive analysis of multiple PE SNP datasets has emphasized the importance of ACVR2A gene variants in the onset of PE, especially in early-onset cases (*Ferreira et al., 2015*; *Yanan et al., 2020*; *Roten et al., 2009*; *Fitzpatrick et al., 2009*; *Amosco et al., 2019*; *Moses et al., 2006*; *Zeybek et al., 2013*; *Glotov et al., 2019*). While this study did not directly assess the impact of ACVR2A SNPs on gene expression, prior research has suggested that SNPs in regulatory regions (e.g., promoters, enhancers, or untranslated regions) can affect transcription factor binding, RNA stability, or splicing efficiency, ultimately altering transcript levels (*Albert and Kruglyak, 2015*). Such mechanisms could explain the observed downregulation of ACVR2A in PE placental samples, as shown in this study. ACVR2A regulates reproductive functions such as decidualization, trophoblastic invasion, and placenta formation, with studies showing abnormal expression in the decidua of PE patients (*Manohar-Sindhu et al., 2023*; *Funghi et al., 2018*; *Yong et al., 2018b*; *Pinyol et al., 2021*; *Monsivais et al., 2021*). Studies have found that ACVR2A is abnormally expressed in the decidua of PE patients, further supporting its crucial role (*Yong et al., 2018b*). The predominant expression of this receptor in endometrial, placental, and vascular endothelial cells underscores its central role in pregnancy-related mechanisms (*Lokki et al., 2011*). Therefore, the present study focused on understanding whether ACVR2A influences the biological behavior of trophoblastic cells during early placental development and how this influence contributes to the development of PE.

To validate the expression of ACVR2A in PE, we initially examined public transcriptomic databases, which revealed significantly lower ACVR2A expression in placental samples from PE patients, consistent with GWAS findings across diverse populations. This was further confirmed by analyzing placental samples from PE patients and normotensive pregnant women. After meticulously minimizing blood cell contamination, we performed western blot, RT-PCR, and immunohistochemistry, all of which consistently demonstrated reduced ACVR2A expression in PE placentas. While age and pre-pregnancy BMI were matched between the PE and control groups, we specifically included only PE patients without comorbidities, such as gestational diabetes or chronic hypertension, to minimize confounding factors. This strict criterion ensures that the observed differences in ACVR2A expression are specifically associated with PE. Future studies with larger cohorts and diverse clinical characteristics are needed to confirm these findings and investigate ACVR2A expression across PE subtypes.

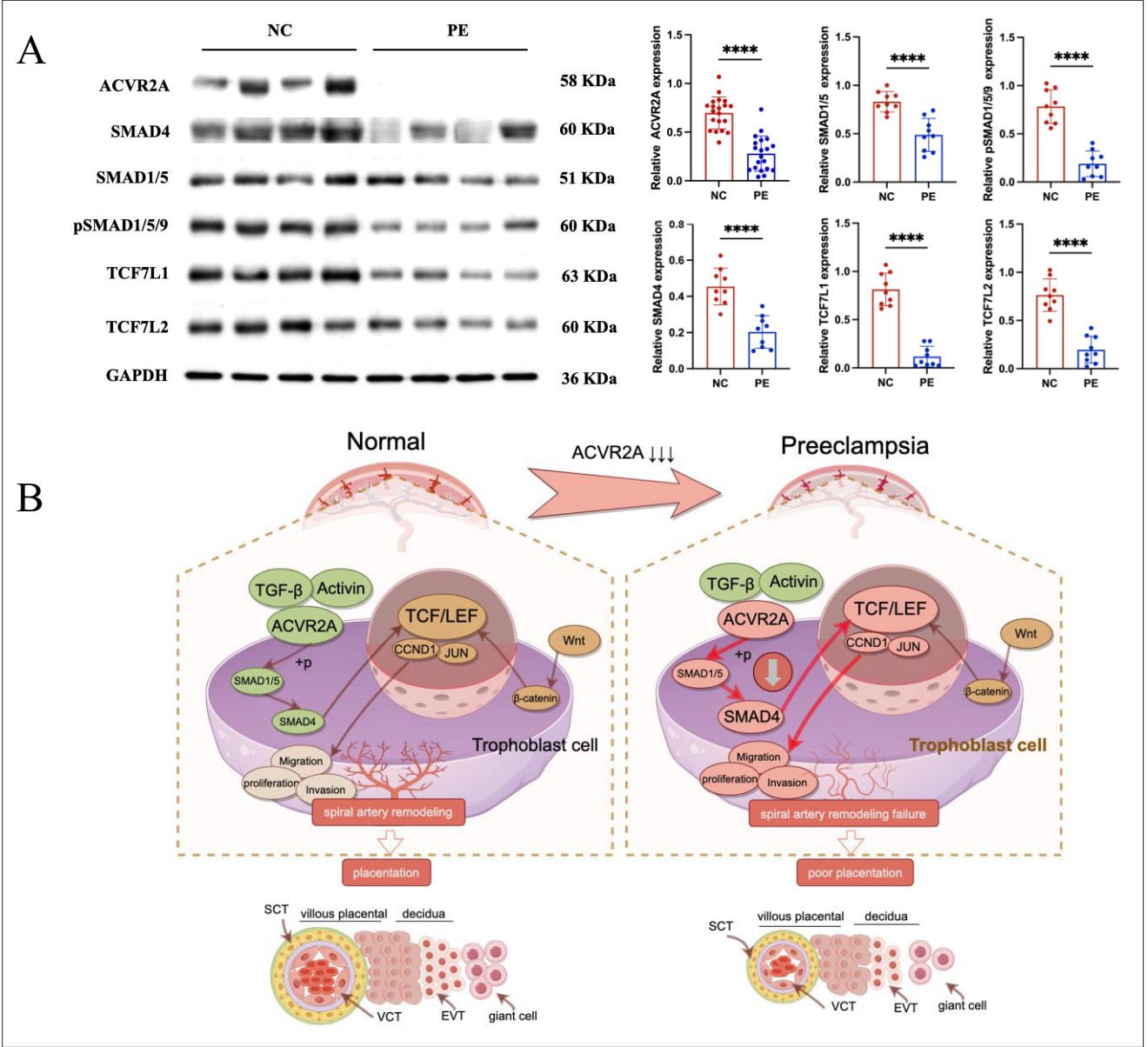

**Figure 7.** Reduced ACVR2A expression impairs SMAD and TCF7/c-JUN signaling in pre-eclampsia, leading to abnormal trophoblast function. (**A**) Western blot analysis showing reduced expression of ACVR2A, SMAD4, SMAD1/5, pSMAD1/5/9, TCF7L1, and TCF7L2 in PE placental tissues compared to normotensive controls. GAPDH was used as a loading control. Densitometric analysis quantifying protein expression levels is shown in the accompanying bar graphs. Data are presented as mean ± SD (*n* = 10 for each group). Statistical significance was determined using a two-tailed Student's *t*-test (****p < 0.0001). (**B**) Schematic illustration of the proposed mechanisms by which ACVR2A regulates trophoblast cell function in normal and pre-eclampsia conditions. (Colors: green indicates normal conditions and functional pathways, while red highlights abnormalities or disrupted pathways in pre-eclampsia. Arrows: solid dark brown arrows: normal signaling pathways and interactions. Bold red arrows: disrupted or abnormal signaling pathways in pre-eclampsia.) In normal placentas, ACVR2A activates the SMAD1/5-SMAD4 axis, promoting trophoblast invasion, migration, and spiral artery remodeling via the TCF7/c-JUN pathway. In pre-eclampsia, reduced ACVR2A expression impairs SMAD signaling, leading to downregulation of TCF7 and its downstream targets (e.g., CCND1 and JUN), contributing to poor placentation. (draw by FigDraw).

The online version of this article includes the following source data for figure 7:

**Source data 1.** Original data related to *Figure 7A* western blot (labelled).

**Source data 2.** Original data related to *Figure 7A* western blot.

**Source data 3.** Original data related to *Figure 7A* western blot.

Interestingly, our findings also highlight that ACVR2A is predominantly localized to the cell surface of trophoblast cells, consistent with its established role as a membrane receptor mediating activin signaling (*Attisano and Wrana, 2002*). However, we observed some intracellular staining, which may reflect receptor trafficking, recycling, or internalization processes during ligand binding. Previous studies have reported that activin receptors, including ACVR2A, can undergo internalization during ligand binding as part of their signaling mechanisms or under specific physiological or pathological conditions (e.g., BMP or Activin A interactions) (*Olsen et al., 2015*). Further investigation into the receptor's trafficking dynamics may help elucidate its functional implications in trophoblast biology.

Studies suggested that despite being distinct cell types, tumor and trophoblast cells exhibit similarities in specific biological behaviors, encompassing cell proliferation and growth, cell migration and invasion, abnormal cell signaling pathways, angiogenesis, and immune system evasion (*Lokki et al., 2011*; *Afzal et al., 2019*; *Perez-Garcia et al., 2021*; *Costanzo et al., 2018*). Dong et al. discovered that circular RNA ACVR2A is implicated in the invasion and migration of bladder cancer cells (*Dong et al., 2019*). Additional studies indicated that the activin A/ACVR2A axis promotes metastasis in colon cancer by selectively activating SMAD2 (*Zhang et al., 2023*; *Zhuo et al., 2018*). ACVR2A and ALK3 regulate endometrial receptivity and embryo implantation in mice, with knockout mice showing significant reproductive defects (*Monsivais et al., 2021*). Ferreira et al. identified a significant association between ACVR2A and early-onset PE (*Ferreira et al., 2015*). These findings underscore the critical role of ACVR2A in cell growth and development, prompting exploration of its contribution to PE through modulation of abnormal trophoblast invasion. The assessment of ACVR2A mRNA expression levels across multiple cell lines revealed that HTR8/SVneo and JAR cells exhibited comparable levels to tumor cells such as A549, suggesting a potential role for ACVR2A in trophoblast cell functions. It is important to note, however, that JAR cells are derived from choriocarcinoma and HTR8/SVneo cells are immortalized, which may affect their behavior and gene expression compared to normal trophoblast cells. While these models are widely used to study trophoblast functions due to their invasive and proliferative properties, their tumorigenic or immortalized origins could influence the observed ACVR2A expression. Future studies using primary trophoblast cells or in vivo models are needed to confirm these findings and further explore ACVR2A's physiological role in placental development.

CRISPR/Cas9, developed by researchers like George Church and Jennifer Doudna, uses guide RNAs to target-specific genomic regions, allowing for precise gene removal, insertion, silencing, or activation, with advantages like complete gene deletion despite potential off-target effects (*Lingor, 2010*; *Schuster et al., 2019*; *Manghwar et al., 2019*). CRISPR/Cas9 technology was used to delete the ACVR2A gene in two trophoblast cell lines to further explore the correlation between PE and ACVR2A. The successful deletion of ACVR2A was confirmed through Sanger sequencing, genotype identification, and RT-PCR. This precise and targeted gene editing allowed validation of the effect of the ACVR2A gene on various trophoblast cell lines. The deletion resulted in suppressed cell growth, impeded migration, and reduced invasion capabilities, highlighting ACVR2A's crucial role in these processes. These findings suggest that ACVR2A plays a significant role in trophoblast cell functions, which are critical for proper placental development. The observed suppression of cell growth, migration, and invasion upon ACVR2A deletion indicates that this gene is integral to the invasive behavior of trophoblast cells, a process essential for normal placentation and successful pregnancy outcomes.

Our findings indicate that ACVR2A plays a pivotal role in trophoblast cell invasion, migration, and proliferation, likely through the TCF7/c-JUN pathway. However, ACVR2A is also a key activator of BMP signaling, particularly through the Smad1/5 axis. A recent study by Monsivais et al. demonstrated that ACVR2A-mediated Smad1/5 signaling is essential for endometrial receptivity and implantation, underscoring the importance of this pathway in reproductive biology (*Monsivais et al., 2021*). Consistent with this, our results revealed significant downregulation of Smad1/5 in PE placentas, suggesting that disruptions in this axis may contribute to the impaired trophoblast function observed in PE. While Smad2/3/4 were not directly assessed in this study, previous research has shown that Smad2/3 can interact with TCF/LEF transcription factors to regulate Wnt-related target genes (*DiRenzo et al., 2016*; *Szilágyi et al., 2022*). This highlights potential cross-talk between TGF-β family signaling and the TCF7/c-JUN axis. In PE, such interactions may be disrupted, further impairing trophoblast invasion and migration. Future studies should aim to investigate the interplay between Smad1/5, Smad2/3/4, and TCF7/c-JUN pathways to better understand the molecular mechanisms underlying PE pathogenesis (*Zhang et al., 2023*; *Guo and Wang, 2009*).

Our findings also face certain limitations that should be acknowledged. The observed reduction in proliferation in ACVR2A knockout cells may partially influence the invasion and migration assay results. Specifically, fewer proliferating cells could inherently lead to fewer invading or migrating cells, complicating the interpretation of these assays. To mitigate this issue, we conducted the invasion and migration assays under low-serum conditions (1–2% serum), as serum-free conditions negatively affected cell viability in preliminary trials (*Pijuan et al., 2019*). Despite these efforts, the impact of reduced proliferation remains a limitation. Future studies could use approaches such as normalizing cell numbers or employing proliferation-independent methods to confirm the specific role of ACVR2A in invasion and migration.

Another potential mechanism for ACVR2A in PE involves its role in the endometrium. Studies in mice have shown that uterine-specific knockout of ACVR2A in progesterone receptor-expressing cells results in embryo implantation failure, underscoring its critical role in uterine receptivity and embryo implantation (*Monsivais et al., 2021*). This suggests that ACVR2A may influence PE not only through its effects on trophoblast function but also via uterine signaling pathways. In this study, we provide novel insights into the role of ACVR2A in trophoblast cell migration and invasion using CRISPR/Cas9 knockout cell lines. While these findings are significant, we acknowledge that the lack of in vivo validation is a limitation. Future studies employing in vivo models, such as trophoblast-specific ACVR2A knockout mice, could provide further evidence of ACVR2A's regulatory effects and its contribution to placental development and PE pathogenesis. Additionally, larger clinical studies are needed to confirm the clinical relevance of ACVR2A expression in diverse patient populations. Addressing these aspects in future research will be essential to fully elucidate the translational potential of our findings.

## Materials and methods
### Collection of placenta and decidua specimens/subject recruitment and placental sampling

This study was approved by the Hospital Ethics Committee (SWYX:NO2021-352). Between October 2022 and October 2024, 20 PE patients and 20 normal pregnant women admitted to the Third Affiliated Hospital of Guangzhou Medical University for cesarean section were randomly selected. The exclusion criteria included gestational diabetes, cardiovascular disease, history of thyroid disease, history of autoimmune disease, history of hypertension, intrahepatic cholestasis of pregnancy, anemia, infectious disease, drug use, recent acute or chronic infectious disease, and incomplete information recording. The control group consisted of 20 non-hypertensive healthy pregnant women who underwent cesarean section due to breech and cicatricial uterus. In accordance with the guidelines from the American College of Obstetricians and Gynecologists (*ACOG Publications, 2020*), PE is diagnosed as hypertension (systolic blood pressure ≥140 mmHg or diastolic blood pressure ≥90 mmHg on at least two occasions) in combination with one or more of the following: proteinuria (≥300 mg/24 hr urine collection or protein/creatinine ratio ≥0.3), thrombocytopenia, elevated serum creatinine, elevated liver enzymes, pulmonary edema, or new-onset headache unresponsive to treatment (*ACOG Publications, 2020*). In this study, we included 40 participants, comprising 20 PE patients and 20 normal pregnancies. To minimize potential confounding factors, only PE patients without any comorbidities, such as gestational diabetes, chronic hypertension, or other pregnancy-related complications, were included. This strict selection criterion was designed to ensure that the observed differences in ACVR2A expression are specifically associated with PE. All patients in the PE group met these diagnostic criteria, and detailed clinical characteristics are provided in *Supplementary file 1*.

All specimens were collected during cesarean sections. Within 5 min after placental delivery, tissue samples measuring approximately 1 × 1 × 1 cm were vertically excised at the junction between the center of the placenta and the umbilical cord. Obvious abnormal regions, such as areas with bleeding, infarction, or calcification, were carefully avoided to ensure sample integrity. The tissue was repeatedly rinsed with sterile saline under aseptic conditions to remove residual blood, then cut into small fragments. The fragments were immediately placed into cryogenic tubes, flash-frozen in liquid nitrogen, and subsequently transferred to a −8°C freezer within 24 hr to preserve RNA and protein integrity for downstream analyses.

## Cell culture

All cell lines used in this study were purchased from established suppliers. HTR8/SVneo cells were obtained from the U.S. Typical Culture Preservation Center, and JAR cells were purchased from Wuhan Punosei Life Technology. Other cell lines, including A549, Huh7, MEG-01, and NCI-H358, were similarly sourced from commercial suppliers. All cell lines were cultured following standard protocols recommended by the suppliers. No other cell contamination was identified by STR. HTR8/SVneo and JAR cells were maintained in RPMI-1640 medium (Gibco) supplemented with 10% fetal bovine serum (FBS, Gibco) and 1% penicillin/streptomycin (Servicebio, China). Cancer cell lines such as A549 and Huh7 were cultured in DMEM or RPMI-1640 medium (Gibco) supplemented with 10% FBS and 1% penicillin/streptomycin. Cells were incubated at 37°C in a humidified atmosphere with 5% $CO_2$, and routine testing for mycoplasma contamination was performed every 2 weeks to ensure the reliability of experimental results.

## CRISPR/Cas9-targeted deletion of ACVR2A

ACVR2A sgRNAs were designed using the online tool provided by the Broad Institute (https://portals.broadinstitute.org/gppx/crispick/public). Two sgRNA sequences were selected based on their high predicted on-target efficiency and low off-target potential (forward: 5'-GAA GGC ACA TCC TGA CTT GT-'; reverse: 5'-GAT GAC ACA ATCTTCTGCAC-3'). Potential off-target sites were further assessed using Cas-OFFinder (http://www.rgenome.net/cas-offinder/). The sgRNAs were synthesized in vitro by transcription and individually combined with a gRNA scaffold to form functional sgRNAs. These sgRNAs were then complexed with Cas9 protein to assemble ribonucleoprotein (RNP) complexes. The RNP complexes were delivered into HTR8/SVneo and JAR cell lines via electroporation using the Celetrix system, following the manufacturer's protocol. The electroporation was performed with an sgRNA–Cas9 mixture in OPTI-MEM buffer. After electroporation, cells were cultured in RPMI 1640 medium containing 10% FBS for 24 hr. Successful incorporation of the RNP complex was confirmed through PCR and sequencing. ACVR2A deletion in single monoclonal cell lines was validated through a multistep process, including Sanger sequencing, agarose gel electrophoresis, and RT-PCR. Primers specifically designed for ACVR2A knockout verification were used during this process (*Supplementary file 4*). This rigorous validation ensures the reliability of the generated knockout cell lines for downstream functional studies.

## RT-qPCR

Total RNA was isolated from cells using a Total RNA Extraction Kit (TIANGEN, China) following the manufacturer's instructions. The extracted RNA was reverse-transcribed into complementary DNA (cDNA) using a cDNA Synthesis Kit (Takara, Japan) according to the protocol provided by the manufacturer. Primers used for quantitative analysis were synthesized by Takara (Japan), and their sequences are listed in *Supplementary file 2*. The mRNA levels were quantified by SYBR Green (Roche, Germany), and dissolution curve analysis was performed to ensure amplification specificity. The PCR protocol consisted of 40 cycles: initial denaturation at 95°C for 10 min, followed by denaturation at 95°C for 5 s, annealing at 63.3°C for 30 s, and extension at 72°C for 10 s. Dissociation curve analysis was performed post-amplification to confirm the specificity of the PCR products. Cycle threshold (Ct) values were used for relative quantification, and all reactions achieved an amplification efficiency between 95% and 100%. These results ensure both the reliability and reproducibility of the quantification process.

## Western blot analysis

Cells were harvested using a cell scraper and resuspended in RIPA buffer supplemented with a protease inhibitor cocktail and phosphatase inhibitor (Sigma-Aldrich, USA). The cell suspension was incubated on ice for 30 min to ensure complete lysis. After incubation, the lysates were centrifuged at 12,000 rpm for 15 min at 4°C to remove cell debris. The supernatant containing protein lysates was collected, and the protein concentration was determined using a BCA protein assay kit (Thermo Fisher Scientific, USA). For tissue samples, placental tissues were homogenized in RIPA buffer containing the same supplements (protease and phosphatase inhibitors). The homogenate was incubated on ice for 30 min and centrifuged at 12,000 rpm for 15 min at 4°C to remove debris. The supernatant containing protein lysates was collected, and protein concentration was similarly determined using

the BCA protein assay kit. Equal amounts of protein (20–30 μg per sample) were separated by SDS–PAGE (12% resolving gel, 5% stacking gel) and electro-transferred onto a PVDF membrane (EMD Millipore, Germany). The membrane was blocked with 5% BSA in TBST (TBS, pH 7.4, 0.2% Tween-20) at room temperature for 1 hr and incubated overnight at 4°C with primary antibodies against ACVR2A (Thermo Fisher Scientific, USA) and GAPDH (Santa Cruz, USA) at specified dilutions. On the following day, the membrane was washed three times with TBST and incubated in darkness for 2 hr with Dylight 680-conjugated secondary antibodies (KPL, USA). Images of the membrane were acquired using a Li-Cor Odyssey Clx Infrared Imaging System (LI-Cor Biotechnology, Lincoln, NE, USA). Densitometry analysis of protein bands was performed using ImageJ software, with GAPDH used as the internal control for normalization.

## IHC staining and data analysis

IHC staining was conducted on 4 μm paraffin sections, which were dried for 30 min at 60°C, dewaxed using an environmentally friendly solution, and gradually dehydrated through an alcohol gradient. Antigen retrieval was performed at 98°C in EDTA (pH 9.0) for 15 min, and endogenous peroxidase activity was quenched with 3% H2O2 at room temperature for 25 min. Sections were incubated over-night at 4°C with primary antibodies at appropriate concentrations (details in *Supplementary file 4*), followed by PBS washing and 50 min incubation with HRP-labeled secondary antibodies at room temperature. The immunoperoxidase signal was visualized using a 3,3'-diaminobenzene (DAB) solution, followed by hematoxylin counterstaining.

For quantification, IHC staining was assessed across the entire tissue section to capture global expression patterns, with representative regions from anchoring villous areas selected for detailed comparison between NC and PE groups. Images were captured under identical magnifications and lighting conditions. The DAB signal (brown, protein-specific) and hematoxylin signal (blue, nuclear staining) were separated using ImageJ's color deconvolution plugin. The DAB signal was quantified as integrated optical density (OD) within the selected regions. To ensure consistency, all samples were processed under identical conditions, including the same antibody dilution, incubation times, and DAB development durations. Negative controls (without primary antibody) were included to monitor background staining, and the DAB reaction was stopped simultaneously for all samples. Quantified signal intensities were normalized to the area of the regions analyzed, and comparisons between NC and PE groups were performed using statistical tests (e.g., *t*-test).

## CCK-8 assay for cell proliferation

The wild-type cells (WT group) and ACVR2A-knockout cells (ACVR2A-KO group) were seeded into 96-well plates at a density of 10,000 cells per well in 100 μl of RPMI 1640 medium supplemented with 10% FBS. The cells were incubated overnight at 37°C in a humidified incubator with 5% CO$_2$. Cell proliferation was assessed using the CCK-8 assay (CAT: G4103-1ML, Servicebio) according to the manufacturer's protocol. At 24, 48, and 72 hr after seeding, 10 μl of CCK-8 reagent was added to each well. After incubating the plates at 37°C for 2 hr the OD of the samples was measured at 450 nm using a microplate reader. The cell growth curve was generated based on the average OD values from three independent biological replicates. Background OD values from wells containing only medium and CCK-8 reagent were subtracted before analysis. Statistical comparisons of proliferation rates between the WT and ACVR2A-KO groups were performed using a Student's *t*-test.

## Wound healing assay

The ability of ACVR2A to enhance cell migration in JAR and HTR8/SVneo cells was evaluated using wound healing (scratch) assays. The cell lines were seeded into 6-well culture plates and cultured until they reached approximately 90% confluence. A sterile 200 μl pipette tip was used to create a single scratch in the cellular monolayer. Detached cells and debris were removed by rinsing with PBS, 1640 medium supplemented with 1% FBS was added to each well to minimize proliferation during the assay. Images of the cells were captured at 0, 24, 48, and 72 hr using a phase-contrast microscope. The migratory capability of the cells was determined at 24, 48, and 72 hr post-scratch by measuring the width of the wound in three randomly selected fields simultaneously. The migration index was computed using the following formula, with T representing time (24, 48, or 72 hr):

$$Healing\ Rate(\%) = (\frac{OH\ scratch\ area - TH\ scratch\ area}{OH\ scratch\ area}) \times 100\%$$

The data represent the mean ± SD of three independent experiments. Statistical significance between groups was determined using a Student's *t*-test.

## Transswell invasion assays

First, the upper chamber of a 12-well Transwell insert with an 8-µm pore size (Corning, USA) was coated with 50 µl of Matrigel (diluted 1:3 in serum-free medium) and allowed to polymerize at 37°C for 30 min. Next, $1.5 \times 10^5$ cells from each cell line were suspended in 200 µl of serum-free medium and seeded into the upper chamber. The lower chamber was filled with 800 µl of growth medium containing 10% FBS to serve as a chemoattractant. The Transwell plates were incubated at 37°C in a humidified incubator with 5% $CO_2$ for 24 hr. After incubation, the non-migrating cells on the upper side of the Transwell membrane were carefully removed with a cotton swab. The cells that had migrated to the bottom surface of the membrane were fixed with 4% paraformaldehyde (PFA) at room temperature for 15 min, then stained with 1% crystal violet for 20 min. Excess dye was removed by washing the inserts with PBS, and the migrated cells were counted under a light microscope in five randomly selected fields per insert. The invading cells were quantified by randomly selecting 5 fields of view per sample under ×300 magnification. Images shown in the figure were captured at lower magnification (×100) to provide a comprehensive visual comparison between experimental and control groups. Each experiment was independently repeated at least 10 times to ensure consistency and reliability of the results.

## Plate cloning formation assay

Colony formation assays were conducted using ACVR2A knockout JAR and HTR8/SVneo cell lines. For each group, the cells were seeded at a density of 1000 cells/well in a 6-well plate (NEST) containing 15% FBS supplemented with 1640 medium and cultured for 14 days. The medium was changed every 3 days. When the majority of cell clones expanded to more than 100 cells, the cells were washed with PBS, fixed with 4% PFA for 20 min, and stained with 1% crystal violet for 20 min at room temperature. The excess crystal violet was washed away with PBS, and the dishes were air dried. The cell clones were manually counted under a dissecting microscope. The statistical data on colony formation were derived from three independent experiments and expressed as mean ± SD.

## Transcriptomic analysis

RNA was extracted from four cell lines (ACVR2A KO-JAR, WT-JAR, ACVR2A KO-HTR8/SVneo, and WT-HTR8/SVneo) using TRIzol reagent (Invitrogen, USA) following the manufacturer's guidelines. RNA quality, purity, and concentration were assessed with a NanoDrop 2000 (Thermo Fisher, USA) and the RNA Nano 6000 Assay Kit (Agilent Bioanalyzer 2100, Agilent, USA). Transcriptomic analysis was conducted by Biomker Technologies (Guangzhou, China). Differential gene expression was analyzed using the DESeq2 R package (version 1.10.1). Benjamini and Hochberg's method was applied to control the false discovery rate (FDR), with genes meeting the thresholds of FDR <0.05 and |log$_2$(fold change) |≥1 considered significant. Pathway enrichment analysis of differentially expressed genes was performed using KOBAS (version 3.0) software.

Public RNA-seq data (GSE114691) from the GEO database, comprising 20 PE and 21 control placental samples, were analyzed to assess ACVR2A expression and identify differentially expressed genes. PE and control groups were defined based on clinical criteria, including hypertension (systolic blood pressure ≥140 mmHg or diastolic blood pressure ≥90 mmHg) and proteinuria (≥300 mg/24 hr urine collection or protein/creatinine ratio ≥0.3). The gestational ages for these samples ranged from 34 to 38 weeks. Differential gene expression analysis was conducted using the DESeq2 R package (version 1.10.1), with thresholds set at FDR <0.05 and |log$_2$(fold change) |≥1. GSEA was performed to identify enriched pathways, with a specific focus on the WNT signaling pathway.

## Statistical analysis

All figures were generated using GraphPad Prism (version 10) software. Data are expressed as the mean ± SD. For data following a normal distribution, parametric tests such as the two-tailed unpaired Student's *t*-test and one-way ANOVA followed by Tukey's post hoc test were used. For comparisons

involving two independent variables, two-way ANOVA was applied with Bonferroni correction for multiple comparisons. For non-normally distributed data, non-parametric tests such as the Mann–Whitney $U$ test were utilized. Normality was assessed using the Shapiro–Wilk test. The suitability of parametric tests was verified by checking the assumptions of normality and homogeneity of variances. When these assumptions were not met, appropriate non-parametric alternatives were used. Specific statistical tests used for each figure and table are detailed in the respective legends and captions. For example, the Mann–Whitney $U$ test was applied in *Supplementary file 2* due to the non-normal distribution of the data. A significance level of $p < 0.05$ was considered statistically significant. All experimental conditions were performed in triplicate unless otherwise specified.

## Acknowledgements

This study was supported by the Guangdong Basic and Applied Basic Research Fund (Guangdong-Shenzhen Joint Fund) (2021B1515120070, 2023A1515010872), the National Key R&D Program of China (2021YFC2701500, 2019YFA0110804), and Guangzhou fundamental research project jointly funded by School (Institution) (high-level university) (202102010131).

## Additional information

### Funding

| Funder | Grant reference number | Author |
|---|---|---|
| Guangdong Basic and Applied Basic Research Fund | 2021B1515120070 | Yi Yang |
| Guangdong Basic and Applied Basic Research Fund | 2023A1515010872 | Yi Yang |
| National Key Research and Development Program of China | 2021YFC2701500 | Fang He |
| Guangzhou Fundamental Research Project jointly funded by School | 202102010131 | Fang He |
| National Key Research and Development Program of China | 2019YFA0110804 | Fang He |

The funders had no role in study design, data collection, and interpretation, or the decision to submit the work for publication.

### Author contributions

Shujing Yang, Data curation, Investigation, Methodology, Writing - original draft, Writing - review and editing; Huanyao Liu, Data curation, Investigation, Writing - original draft, Writing - review and editing; Jieshi Hu, Data curation, Formal analysis, Investigation; Binjun Chen, Data curation, Investigation; Wanlu An, Xuwen Song, Investigation; Yi Yang, Conceptualization, Resources, Supervision, Funding acquisition, Visualization, Methodology, Project administration; Fang He, Conceptualization, Resources, Data curation, Formal analysis, Supervision, Funding acquisition, Visualization, Project administration

### Author ORCIDs

Fang He  https://orcid.org/0000-0003-3362-6002

### Ethics

This study was approved by the Hospital Ethics Committee (SWYX:NO2021-352). All participants in this study provided a written informed consent for donating blood. The Ethics Committee of The Third Affiliated Hospital of Guangzhou Medical University (Guangzhou, China) approved the experiments

using human cells. The animal experiments were approved by The Third Affiliated Hospital of Guangzhou Medical University. All methods were performed in accordance with the approved guidelines.

Reviewer #1 (Public review): https://doi.org/10.7554/eLife.101236.3.sa1
Reviewer #2 (Public review): https://doi.org/10.7554/eLife.101236.3.sa2
Author response https://doi.org/10.7554/eLife.101236.3.sa3

---

## Additional files

### Supplementary files
Supplementary file 1. Clinical characteristics of pre-eclamptic and normal control pregnancies.
Supplementary file 2. Sequences of the primers for RT-qPCR.
Supplementary file 3. Sequences of the genotyping primers.
Supplementary file 4. Details of immunohistochemical antibodies.
MDAR checklist

### Data availability
Sequencing data is available from the NCBI database, PRJNA1250493.

The following dataset was generated:

| Author(s) | Year | Dataset title | Dataset URL | Database and Identifier |
|---|---|---|---|---|
| He F | 2024 | ACVR2A Facilitates Trophoblast Cell Invasion through TCF7/c-JUN Pathway in Pre-eclampsia Progression | https://www.ncbi.nlm.nih.gov/bioproject/PRJNA1250493/ | NCBI BioProject, PRJNA1250493 |

The following previously published dataset was used:

| Author(s) | Year | Dataset title | Dataset URL | Database and Identifier |
|---|---|---|---|---|
| Awamleh Z, Gloor GB, Han V | 2019 | Placental transcriptome in pregnancies complicated by Intrauterine growth restriction (IUGR) and preeclampsia (PE) | https://www.ncbi.nlm.nih.gov/geo/query/acc.cgi?acc=GSE114691 | NCBI Gene Expression Omnibus, GSE114691 |

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
