## [Editor Report · eLife Assessment]

The role of ACVR2A is potentially of importance to both the biology of trophoblast cells and to the pathogenesis of preeclampsia. In this manuscript, the authors have taken a **useful** first step towards better understanding this protein using a loss of function model in trophoblast cell lines and then examining invasion, proliferation, and transcription in these cells. The study is **solid** and further in vivo evidence on how target factors participate in the occurrence of placental structural disorders and diseases through potential downstream pathways will be invaluable in the future.

---

## [Referee Report · Reviewer #1 (Public review)]

Summary:

This study has preliminarily revealed the role of ACVR2A in trophoblast cell function, including its effects on migration, invasion, proliferation, and clonal formation, as well as its downstream signaling pathways.

Strengths:

The use of multiple experimental techniques, such as CRISPR/Cas9-mediated gene knockout, RNA-seq, and functional assays (e.g., Transwell, colony formation, and scratch assays), is commendable and demonstrates the authors' effort to elucidate the molecular mechanisms underlying ACVR2A's regulation of trophoblast function. The RNA-seq analysis and subsequent GSEA findings offer valuable insights into the pathways affected by ACVR2A knockout, particularly the Wnt and TCF7/c-JUN signaling pathways.

Weaknesses:

The current findings provide valuable insights into the role of ACVR2A in trophoblast cell function and its involvement in the regulation of migration, invasion, and proliferation, further validation in both in vitro and in vivo models would strengthen the conclusions. Additional techniques, such as animal models and more advanced clinical sample analyses, would help strengthen the conclusions and provide a more comprehensive understanding of the molecular pathways involved.

---

## [Referee Report · Reviewer #2 (Public review)]

Summary:

ACVR2A is one of a handful of genes for which significant correlations between associated SNPs and the incidences of preeclampsia have been found in multiple populations. It is one of the TGFB family receptors, and multiple ligands of ACVR2A, as well as its coreceptors and related inhibitors, have been implicated in placental development, trophoblast invasion, and embryo implantation. This useful study builds on this knowledge by showing that ACVR2A knockout in trophoblast-related cell lines reduces trophoblast invasion, which could tie together many of these observations. The implication of cross-talk between the WNT and ACRV2A/SMAD2 pathways is an important contribution to the understanding of the regulation of trophoblast function.

Strengths:

(1) ACVR2A is one of very few genes implicated in preeclampsia in multiple human populations, yet its role in pathogenesis is not very well studied and this study begins to address that hole in our knowledge.

(2) ACVR2A is also indirectly implicated in trophoblast invasion and trophoblast development via its connections to many ligands, inhibitors, and coreceptors, suggesting its potential importance.

(3) The authors have used multiple cell lines to verify their most important observations.

Editors' note: Following the first round of peer review, the original reviewers were not available to review the revised manuscript. As several specific weakness detailed by the reviewers were largely addressed in the revised manuscript, they are not included here.

---

## [Author Response]

The following is the authors’ response to the original reviews

**Public Reviews:**

**Reviewer #1 (Public review):**
Summary:This study has preliminarily revealed the role of ACVR2A in trophoblast cell function, including its effects on migration, invasion, proliferation, and clonal formation, as well as its downstream signaling pathways.Strengths:The use of multiple experimental techniques, such as CRISPR/Cas9-mediated gene knockout, RNA-seq, and functional assays (e.g., Transwell, colony formation, and scratch assays), is commendable and demonstrates the authors' effort to elucidate the molecular mechanisms underlying ACVR2A's regulation of trophoblast function. The RNA-seq analysis and subsequent GSEA findings offer valuable insights into the pathways affected by ACVR2A knockout, particularly the Wnt and TCF7/c-JUN signaling pathways.Weaknesses:The molecular mechanisms underlying this study require further exploration through additional experiments. While the current findings provide valuable insights into the role of ACVR2A in trophoblast cell function and its involvement in the regulation of migration, invasion, and proliferation, further validation in both in vitro and in vivo models is needed. Additionally, more experiments are required to establish the functional relevance of the TCF7/c-JUN pathway and its clinical significance, particularly in relation to pre-eclampsia. Additional techniques, such as animal models and more advanced clinical sample analyses, would help strengthen the conclusions and provide a more comprehensive understanding of the molecular pathways involved.
**Reviewer #2 (Public review):**
Summary:ACVR2A is one of a handful of genes for which significant correlations between associated SNPs and the incidences of preeclampsia have been found in multiple populations. It is one of the TGFB family receptors, and multiple ligands of ACVR2A, as well as its coreceptors and related inhibitors, have been implicated in placental development, trophoblast invasion, and embryo implantation. This useful study builds on this knowledge by showing that ACVR2A knockout in trophoblast-related cell lines reduces trophoblast invasion, which could tie together many of these observations. Support for this finding is incomplete, as reduced proliferation may be influencing the invasion results. The implication of cross-talk between the WNT and ACRV2A/SMAD2 pathways is an important contribution to the understanding of the regulation of trophoblast function.Strengths:(1) ACVR2A is one of very few genes implicated in preeclampsia in multiple human populations, yet its role in pathogenesis is not very well studied and this study begins to address that hole in our knowledge.(2) ACVR2A is also indirectly implicated in trophoblast invasion and trophoblast development via its connections to many ligands, inhibitors, and coreceptors, suggesting its potential importance.(3) The authors have used multiple cell lines to verify their most important observations.Weaknesses:(1) There are a number of claims made in the introduction without attribution. For example, there are no citations for the claims that family history is a significant risk factor for PE, that inadequate trophoblast invasion of spiral arteries is a key factor, and that immune responses, and reninangiotensin activity are involved.

Thank you for pointing out the lack of citations in some parts of the introduction. We have revised the manuscript to include appropriate references for the claims regarding family history as a risk factor for PE, the role of inadequate trophoblast invasion in spiral arteries, and the involvement of immune responses and the renin-angiotensin system. The revised text now includes citations to well-established studies in the field (Salonen Ros et al., 2000; Chappell LC et al., 2021; Brosens et al., 2002; Knofler et al., 2019; Redman CWG et al., 1999; LaMarca B et al., 2008). We believe these additions improve the scientific rigor of the manuscript.

(2) The introduction states "As a receptor for activin A, ACVR2A..." It's important to acknowledge that ACVR2A is also the receptor for other TGFB family members, with varying affinities and coreceptors. Several TGFB family members are known to regulate trophoblast differentiation and invasion. For example, BMP2 likely stimulates trophoblast invasion at least in part via ACVR2A (PMID 29846546).

Thank you for highlighting the broader role of ACVR2A as a receptor for multiple members of the TGF-β superfamily. We have revised the introduction to acknowledge that ACVR2A is not only the receptor for activin A but also interacts with other ligands, such as BMP2, which likely stimulates trophoblast invasion via ACVR2A (PMID: 29846546). This addition provides a more comprehensive view of ACVR2A's function in trophoblast biology. While the focus of our current study is on activin A, we agree that ACVR2A's role in mediating the effects of other TGF-β family members is an important topic for future research.

(3) An alternative hypothesis for the potential role of ACVR2A in preeclampsia is its functions in the endometrium. In the mouse ACVR2A knockout in the uterus (and other progesterone receptorexpressing cells) leads to embryo implantation failure.

Thank you for bringing up the potential role of ACVR2A in the endometrium as an alternative hypothesis. We have revised the discussion to acknowledge this possibility and cited relevant studies showing that uterine-specific knockout of ACVR2A in mice leads to embryo implantation failure (Monsivais et al., 2021). This suggests that ACVR2A may play a critical role in uterine receptivity and embryo implantation, which could influence placental development and preeclampsia pathogenesis. While our current study focuses on trophoblast-related functions of ACVR2A, we agree that investigating its role in the uterine environment is an important direction for future research.

(4) In the description of the patient population for placental sample collections, preeclampsia is defined only by hypertension, and this is described as being in accordance with ACOG guidelines. ACOG requires a finding of hypertension in combination with either proteinuria or one of the following: thrombocytopenia, elevated creatinine, elevated liver enzymes, pulmonary, edema, and new onset unresponsive headache.

We appreciate the reviewer’s detailed observation regarding the definition of preeclampsia.

We have reviewed and clarified our description of the diagnostic criteria based on the American College of Obstetricians and Gynecologists (ACOG) guidelines. Specifically, we have revised the definition in the Materials and Methods section under "Collection of Placenta and Decidua Specimens," as follows: In accordance with the guidelines from the American College of Obstetricians and Gynecologists (ACOG, 2023), preeclampsia (PE) is diagnosed as hypertension (systolic blood pressure ≥140 mmHg or diastolic blood pressure ≥90 mmHg on at least two occasions) in combination with one or more of the following: proteinuria (≥300 mg/24-hour urine collection or protein/creatinine ratio ≥0.3), thrombocytopenia, elevated serum creatinine, elevated liver enzymes, pulmonary edema, or new-onset headache unresponsive to treatment.

(5) I believe that Figures 1a and 1b are data from a previously published RNAseq dataset, though it is not entirely clear in the text. The methods section does not include a description of the analysis of these data undertaken here. It would be helpful to include at least a brief description of the study these data are taken from - how many samples, how were the PE/control groups defined, gestational age range, where is it from, etc. For the heatmap presented in B, what is the significance of the other genes/ why are they being shown? If the purpose of these two panels is to show differential expression specifically of ACVR2A in this dataset, that could be shown more directly.

Clarification of RNAseq dataset: The Methods section has been revised to specify the dataset source (GEO accession number: GSE114691), which includes 20 PE and 21 control placental samples with gestational ages ranging from 34 to 38 weeks. PE and control groups were defined using clinical criteria such as hypertension and proteinuria, and these details have also been added to the Results section. RNAseq analysis description: We have included details of the differential gene expression analysis in the Methods section. Specifically, the DESeq2 R package was used, with thresholds of FDR < 0.05 and |log2(fold change) | ≥ 1. The selection of WNT pathwayrelated genes in Figure 1B is based on these analyses. Significance of the heatmap genes: The genes displayed in Figure 1B were selected based on their significant differential expression and enrichment in pathways relevant to PE pathogenesis, such as the WNT signaling pathway. We have clarified this in the Results section and updated the figure legend to explain their biological relevance. Purpose of Figures 1A and 1B: Figure 1A emphasizes the downregulation of ACVR2A in PE placentas, while Figure 1B complements this by presenting differentially expressed genes associated with the WNT pathway. These figures collectively highlight the role of ACVR2A in PE and its connection to broader molecular pathways. Text descriptions have been updated to improve clarity and focus.

(6) More information is needed in the methods section to understand how the immunohistochemistry was quantified. "Quantitation was performed" is all that is provided. Was staining quantified across the whole image or only in anchoring villous areas? How were HRP & hematoxylin signals distinguished in ImageJ? How was the overall level of HRP/DAB development kept constant between the NC and PE groups?

Thank you for pointing out the need for more details regarding the quantification of immunohistochemistry (IHC). We have now clarified and expanded the description of the IHC quantification process in the Methods section as follows: Quantification Across the Entire Section: IHC staining was assessed across the entire tissue section to account for global expression patterns. For quantitative analysis, representative regions from the anchoring villous areas, where ACVR2A expression is most prominent, were selected for comparison between NC and PE groups. This ensured that the analysis focused on biologically relevant regions. ImageJ Analysis:

Images of stained sections were captured under identical magnifications and lighting conditions. Hematoxylin (blue, nuclear staining) and DAB/HRP (brown, protein-specific signal) were distinguished using ImageJ's color deconvolution plugin. The DAB/HRP signal was isolated and quantified based on the integrated optical density (IOD) within the selected regions. Consistency in HRP/DAB Development: To maintain consistency between NC and PE groups, all tissue samples were processed under identical experimental conditions, including the same antibody dilution, incubation times, and DAB/HRP development durations. Negative controls (without primary antibody) were included to monitor background staining, and the DAB reaction was stopped simultaneously across all samples to avoid overdevelopment. Statistical Analysis: The quantified DAB signal intensity was normalized to the area of the selected regions, and comparisons between NC and PE groups were performed using statistical tests (e.g., Student’s ttest). Results are reported as mean ± SD. We hope this additional detail addresses your concerns.

(7) In Figure 1E it is not immediately obvious to many readers where the EVT are. It is probably worth circling or putting an arrow to the little region of ACVR2A+ EVT that is shown in the higher magnification image in Figure 1E. These are actually easier to see in the pictures provided in the supplement Figure 1. Of note, the STB is also staining positive. This is worth pointing out in the results text.

Thank you for your suggestion regarding Figure 1E. To make the location of the ACVR2A+ extravillous trophoblasts (EVTs) more apparent, we have updated Figure 1E by adding arrows to indicate the regions of EVTs in the higher magnification image. Additionally, we have included annotations in the supplemental Figure S1 to further aid visualization. We appreciate your observation that syncytiotrophoblasts (STBs) also show positive staining for ACVR2A. We have revised the Results section to explicitly mention this finding and its potential significance.

(8) It is not possible to judge whether the IF images in 1F actually depict anchoring villi. The DAPI is really faint, and it's high magnification, so there isn't a lot of context. Would it be possible to include a lower magnification image that shows where these cells are located within a placental section? It is also somewhat surprising that this receptor is expressed in the cytoplasm rather than at the cell surface. How do the authors explain this?

Thank you for your suggestion to provide more context for the immunofluorescence (IF) images in Figure 1F. To address this, we have included lower magnification images in Supplementary Figure S2, showing the overall structure of the placental section and the location of the anchoring villi. These images help to contextualize the regions analyzed in Figure 1F, which were selected to clearly illustrate ACVR2A expression in extravillous trophoblasts (EVTs). In Figure 1F, we have focused on higher magnification images for better visualization of ACVR2A staining patterns in EVTs. Regarding the subcellular localization of ACVR2A, the receptor is predominantly expressed on the cell surface, as shown in our images. However, some intracellular staining is also observed, which may reflect receptor trafficking or recycling processes, consistent with the behavior of other activin receptors under physiological or pathological conditions. We have clarified these points in the Results and Discussion sections.

(9) The results text makes it sound like the data in Figure 2A are from NCBI & Protein atlas, but the legend says it is qPCR from this lab. The methods do not detail how these various cell lines were grown; only HTR-SVNeo cell culture is described. Similarly, JAR cells are used for several experiments and their culture is not described.

Thank you for pointing out the need for clarification regarding Figure 2A and cell culture methods. The data in Figure 2A were generated using RT-qPCR conducted in our laboratory, not solely based on data from NCBI or the Human Protein Atlas. We have revised the Results section to reflect this more accurately. Regarding the culture conditions, we acknowledge that the methods for other cell lines were not explicitly detailed. For this study, all cell lines, including JAR and other cancer cell lines, were cultured following standard protocols provided by the suppliers. Specifically, JAR cells and other cell lines were purchased from Wuhan Punosei Life Technology and were maintained in RPMI-1640 medium supplemented with 10% fetal bovine serum (FBS) and 1% penicillin/streptomycin under standard conditions (37°C, 5% CO_2_). This information has been added to the Methods section for clarity.

(10) Under RT-qPCR methods, the phrase "cDNA reverse transcription cell RNA was isolated..." does not make any sense.

Thank you for pointing out the unclear phrasing in the RT-qPCR methods section. We agree that the original description was not precise. To address this, we have revised the relevant section to improve clarity and accuracy. Specifically, the methods now explicitly describe the two key steps: RNA isolation and cDNA synthesis. The revised text reads: Total RNA was isolated from cells using a Total RNA Extraction Kit (TIANGEN, China) following the manufacturer’s instructions. The extracted RNA was reverse-transcribed into complementary DNA (cDNA) using a cDNA Synthesis Kit (Takara, Japan) according to the protocol provided by the manufacturer.

(11) The paragraph beginning "Consequently, a potential association..." is quite confusing. It mentions analyzing ACVR2A expression in placentas, but then doesn't point to any results of this kind and repeats describing the results in Figure 2a, from various cell lines.

Thank you for your comment regarding the paragraph beginning with "Consequently, a potential association...". We understand that the current wording may create confusion. The primary aim of this section is to compare ACVR2A expression levels across various cell lines, including trophoblast-derived and non-trophoblast cell lines, to highlight the relevance of ACVR2A in trophoblast function, particularly in invasion and migration. To address your concerns, we have revised the paragraph for clarity and logical flow. The updated text explicitly focuses on the comparison of ACVR2A expression across cell lines (Figure 2A) and how this supports the hypothesis that ACVR2A plays a key role in trophoblast invasion and migration. Additionally, the discussion of placental samples has been separated to avoid confusion with cell line results. We hope this revision resolves the issue.

(12) The authors should acknowledge that the effect of the ACVR2A knockout on proliferation makes it difficult to draw any conclusions from the trophoblast invasion assays. That is, there might be fewer migrating or invading cells in the knockout lines because there are fewer cells, not because the cells that are there are less invasive. Since this is a central conclusion of the study, it is a major drawback.

Thank you for highlighting this important point. We agree that the reduced proliferation observed in ACVR2A knockout cells could influence the results of the invasion assays, as fewer cells may inherently lead to reduced invasion. To minimize this effect, we conducted the invasion and migration assays under low-serum conditions (1–2% serum) to limit cell proliferation during the experimental timeframe. This approach was based on optimization trials and existing literature, as serum-free conditions were found to negatively impact cell viability and experimental reproducibility. While these efforts helped to mitigate the impact of proliferation on the results, we acknowledge this as a limitation of our study and have added this discussion to the manuscript. Future studies could incorporate approaches such as normalizing cell numbers or using additional proliferation-independent methods to confirm the findings. We hope this clarification and the steps taken address your concerns.

(13) The legend and the methods section do not agree on how many fields were selected for counting in the transwell invasion assays in Figure 3C. The methods section and the graph do not match the number of replicate experiments in Figure 3D (the number of replicate experiments isn't described for 3C).

Thank you for pointing out the inconsistencies regarding the number of fields counted and the number of replicates in the Transwell invasion assays (Figure 3C) and colony formation assays (Figure 3D). We apologize for the lack of clarity in the Methods section and figure legend. To address this, we have revised both the figure legends and the Methods section for consistency and added detailed descriptions. For Figure 3C, cell invasion was quantified by randomly selecting 5 fields of view per sample under 300× magnification. Images shown in the figure were taken at lower magnification to provide a better visual comparison between experimental and control groups. For Figure 3D, each experiment was independently repeated at least 10 times to ensure robust and reproducible results. These clarifications have been incorporated into the revised manuscript. We appreciate your feedback and believe this revision improves the clarity and transparency of our methods.

(14) Discussion says "Transcriptome sequencing analysis revealed low ACVR2A expression in placental samples from PE patients, consistent with GWAS results across diverse populations." The authors should explain this briefly. Why would SNPs in ACVR2A necessarily affect levels of the transcript?

Thank you for raising this important point. We acknowledge that our study did not directly investigate how SNPs in the ACVR2A gene affect transcript levels. However, prior studies have suggested that SNPs can influence gene expression through various mechanisms. For example, SNPs in regulatory regions (e.g., promoters, enhancers, or untranslated regions) may affect transcription factor binding, RNA stability, or splicing efficiency, ultimately altering transcript levels. While we did not directly assess the functional consequences of ACVR2A SNPs in this study, the observed downregulation of ACVR2A in PE placentas aligns with the potential regulatory impact of SNPs previously identified in GWAS studies. To address this, we have revised the Discussion section to clarify the relationship between SNPs and transcript levels and acknowledge this limitation.

(15) "The expression levels of ACVR2A mRNA were comparable to those of tumor cells such as A549. This discovery suggested a potential pivotal role of ACVR2A in the biological functions of trophoblast cells, especially in the nurturing layer." Alternatively, ACVR2A expression resembles that of tumors because the cell lines used here are tumor cells (JAR) or immortalized cells (HTR8). These lines are widely used to study trophoblast properties, but the discussion should at least acknowledge the possibility that the behavior of these cells does not always resemble normal trophoblasts.

Thank you for pointing out this important limitation. We agree that the JAR and HTR8/SVneo cell lines, being tumor-derived or immortalized, may not fully replicate the behavior of normal trophoblast cells. While these cell lines are widely used as models for studying trophoblast properties due to their ease of culture and invasive behavior, their gene expression and signaling pathways could partially reflect their tumorigenic or immortalized origins. We have revised the Discussion section to acknowledge this limitation and clarify the interpretation of ACVR2A expression levels in these cells.

(16) The authors should discuss some of what is known about the relationship between the TCF7/c-JUN pathway and the major signaling pathway activated by ACVR2A, Smad 2/3/4. The Wnt and TGFB family cross-talk is quite complex and it has been studied in other systems.

Thank you for highlighting the relationship between the TCF7/c-JUN pathway and Smad2/3/4 signaling. In our study, we chose to focus on Smad1/5 due to its strong association with ACVR2A in placental development, as demonstrated in a recent study(DOI: 10.1038/s41467-021-23571-5). This study showed that the BMP signaling pathway, mediated through ACVR2A-Smad1/5, is essential for endometrial receptivity and embryo implantation. While Smad2/3/4 are wellestablished mediators of TGF-β signaling, Smad1/5 activation is more directly linked to ACVR2A in the context of reproductive biology.

In PE placentas, we observed a significant downregulation of Smad1/5 expression, which supports the hypothesis that ACVR2A-mediated Smad signaling is disrupted in this condition. Although we did not directly assess Smad2/3/4 in this study, prior research has shown that Smad2/3 can interact with TCF/LEF transcription factors to regulate Wnt-related target genes, suggesting potential cross-talk between these pathways. We have now clarified this rationale and included a discussion of these interactions in the revised manuscript.

**Recommendations for the authors:**

**Reviewer #1 (Recommendations for the authors):**
Several points need to be addressed to improve the clarity and robustness of the presented findings:(1) From a clinical perspective, several concerns arise regarding the interpretation of these findings. First, the small sample size of 20 patients may not be representative of the broader population, limiting the generalizability of the results. Additionally, although no significant differences in age and pre-pregnancy BMI were observed between the PE and normal control groups, other clinical variables, such as hypertension or gestational diabetes, may also influence ACVR2A expression and contribute to PE development. Furthermore, while the study suggests a correlation between reduced ACVR2A expression and PE, it remains unclear whether this association holds true across different subtypes of PE or whether there are other underlying clinical factors that could account for these changes in gene expression. These factors need to be considered in future studies to better understand the clinical relevance of ACVR2A in PE.

Thank you for raising these insightful concerns about the clinical interpretation of our findings. We agree that the small sample size of 20 patients may limit the generalizability of our results. To address this, we are actively expanding our cohort by collecting additional clinical samples from PE patients and normotensive controls. This effort aims to strengthen the robustness of our findings and provide stronger evidence for the role of ACVR2A in PE. We would also like to clarify that, during the initial sample collection, we specifically included only PE patients without comorbidities such as gestational diabetes, chronic hypertension, or other pregnancy-related complications. This strict selection criterion was implemented to minimize the potential influence of confounding clinical variables and ensure that our findings specifically reflect the association between ACVR2A expression and PE. While our study provides important initial insights, we recognize the need for larger-scale studies to validate these findings. The ongoing collection of clinical samples will allow us to address this limitation and enhance the translational relevance of our research. We have revised the manuscript to reflect these points and highlight our plans to strengthen the study by increasing the sample size.

(2) The section "Precision Genome Surgery: ACVR2A Knockout via CRISPR/Cas9" in the results contains some issues with expression details. The results section should be more structured, with data presented in a more detailed and clear manner, ensuring that there is a clear connection between each experimental step and its corresponding result. For example, the sentence "Following multiple rounds of monoclonal culture, genotype identification, RT-qPCR and Western blotting (WB) analysis for screening, specific double-knockout monoclonal cell lines were distinctly chosen" contains redundant phrasing and unnecessary details, which affect the flow of the text.

Thank you for your constructive feedback on the “Precision Genome Surgery: ACVR2A Knockout via CRISPR/Cas9” section. We agree that this section can be better structured to present the data in a more detailed and coherent manner. To address this, we have reorganized the results into distinct steps, ensuring a clear connection between each experimental step and its corresponding result. Redundant phrasing has been removed to improve the flow and readability of the text. The revised section emphasizes the purpose of each step, the screening process, and the specific results obtained.

(3) The figure legends and panel labels in Figure 3 should be revised to ensure clarity and consistency. The figure legend should specify the exact panels (e.g., Figure 3A, 3B, 3C, etc.) and clearly describe the experimental conditions and results shown in each part.

Thank you for pointing out the need for improved clarity and consistency in the figure legends and panel labels for Figure 3. We have revised the figure legend to specify each panel (e.g., Figure 3A, 3B, 3C, etc.) and included detailed descriptions of the experimental conditions and results displayed in each part. These updates aim to ensure better understanding and alignment between the figure legend and the panels.

(4) Lack of In Vivo Validation of ACVR2A Knockout: The study does not include in vivo experiments to validate the effects of ACVR2A knockout. It would be important to investigate whether similar regulatory effects of ACVR2A on trophoblast cell migration and invasion can be observed in animal models or in larger clinical studies. The lack of in vivo data raises questions about the translational relevance of the findings.

Thank you for highlighting the importance of in vivo validation to assess the translational relevance of our findings. While we acknowledge that in vivo experiments could provide additional insights into the role of ACVR2A in trophoblast migration and invasion, this study was primarily designed as an in vitro investigation to explore the molecular mechanisms underlying ACVR2A function in trophoblast cells. The choice of an in vitro model allowed us to perform precise and controlled mechanistic analyses, which are critical for establishing a foundation for future research. We agree that in vivo studies using animal models or larger clinical cohorts are important next steps to validate the regulatory effects of ACVR2A on trophoblast function and its contribution to PE pathogenesis. These directions will be pursued in future research to further establish the translational potential of our findings. We have included this perspective in the revised Discussion section.

(5) TCF7/c-JUN Pathway in Clinical Samples: In the study of the TCF7/c-JUN pathway, the authors mention assessing protein expression in clinical samples through immunohistochemistry (IHC). However, the manuscript does not provide a clear explanation of how the findings from laboratory cell models (such as HTR8/SVneo and JAR) relate to the clinical samples. Specifically, while ACVR2A knockout is shown to affect these proteins at the cellular level, it is unclear whether this effect is observed in clinical samples. Therefore, further validation of the TCF7/c-JUN pathway in the cell models and exploration of its relationship with protein expression in clinical samples is necessary. Additional experiments, such as immunofluorescence staining or mass spectrometry, could further confirm the role of the TCF7/c-JUN pathway in cells and provide a more direct comparison with clinical data.

Thank you for highlighting the need to connect findings from cell models to clinical samples, particularly with respect to the TCF7/c-JUN pathway. In response to your comment, we conducted additional experiments using Western blot analysis to evaluate the expression of ACVR2A, SMAD1/5, SMAD4, pSMAD1/5/9, and TCF7L1/TCF7L2 in PE placental tissues compared to normotensive controls (Figure 7A). The results demonstrated significantly reduced expression of these proteins in PE placentas, providing evidence that disruptions in the ACVR2A-SMAD and TCF7/c-JUN signaling pathways observed in vitro are also present in clinical samples.

These findings strengthen the translational relevance of our study by directly linking the molecular mechanisms identified in cell models to clinical observations. We have updated the Results and Discussion sections to incorporate these new data, and we believe this addition addresses your concern about the relationship between in vitro and clinical findings.